# Dynamic Adaptation of Heart Rate and Autonomic Regulation During Training and Recovery Periods in Response to a 12-Week Structured Exercise Programme in Untrained Adult and Geriatric Horses

**DOI:** 10.3390/ani15081122

**Published:** 2025-04-13

**Authors:** Thita Wonghanchao, Kanokpan Sanigavatee, Chanoknun Poochipakorn, Onjira Huangsaksri, Metha Chanda

**Affiliations:** 1Veterinary Clinical Studies Program, Faculty of Veterinary Medicine, Kasetsart University, Kamphaeng Saen Campus, Nakorn Pathom 73140, Thailand; thita.wo@ku.th (T.W.); ksanigavatee@gmail.com (K.S.); 2Center for Veterinary Research and Innovation, Faculty of Veterinary Medicine, Kasetsart University, Bang Khen Campus, Bangkok 10900, Thailand; chanoknun.p@ku.th; 3Department of Large Animal and Wildlife Clinical Science, Faculty of Veterinary Medicine, Kasetsart University, Kamphaeng Saen Campus, Nakorn Pathom 73140, Thailand

**Keywords:** adult, aerobic training, geriatric, heart rate variability, horse, training effect, recovery, welfare

## Abstract

Changes in resting heart rate (HR) and HR variability (HRV) are commonly used to assess stress responses and training effects in horses. However, these measures may not fully capture the outcomes, especially in horses of different ages. This study investigated how HR and HRV changed during 54 min exercise bouts comprising three exercise sessions and recovery periods in untrained adult horses (UAHs) and geriatric horses (UGHs) over a 12-week structured exercise programme. Its findings revealed decreases in the duration of elevated HR, sympathetic nervous system activity, and stress indices during the exercise sessions. HR also consistently decreased over the 12 weeks, indicating improved physical fitness and reduced stress in both UAHs and UGHs after completing the exercise programme. Moreover, HRV returned to baseline faster in UAHs than in UGHs, suggesting quicker physiological adaptation. In conclusion, the 12-week structured exercise programme positively influenced HR and HRV adaptation in both UAHs and UGHs, with the adaptations differing between them. The measurements taken during the exercise sessions and recovery periods may serve as valuable indicators for assessing and distinguishing training effects in horses participating in similar exercise programmes.

## 1. Introduction

Exercise training serves several constructive purposes for horses, enhancing overall performance and well-being [1,2]. Its primary aims include improving or maintaining optimal sports performance, delaying fatigue, minimising injury risk, and encouraging the horse’s enthusiasm for exercise [3,4]. Aerobic training plays a particularly crucial role in positively impacting biological systems by enhancing cardiovascular function [5], lowering resting heart rate (HR) [6,7,8,9,10], and increasing the aerobic capacity of skeletal muscle [11,12,13]. These improvements collectively foster better fitness levels in horses [14,15]. Aerobic training is also crucial in maintaining muscle mass and functional properties of skeletal muscle in aged horses [15,16]. Exercise training can effectively modulate autonomic regulation, as indicated by changes in HR variability (HRV) [9,17,18,19,20,21]. HRV is the variation in time intervals between adjacent heartbeats due to the rhythmic oscillation of the sympathetic (SNS) and parasympathetic (vagal; PNS) nervous systems [22,23]. It reflects an animal’s adaptation capacity and biological flexibility following stress conditions [22,23,24].

Changes in various HRV metrics can indicate specific autonomic nervous system activities. For example, the root mean square of the differences between successive RR intervals (RMSSD), the high-frequency (HF) band, and the standard deviation (SD) of the Poincaré plot perpendicular to the line of identity (SD1) reflect short-term variations and demonstrate PNS dominance [22,23,25]. In contrast, alterations in the SD of normal-to-normal RR intervals (SDNN), the low-frequency (LF) band, and the SD of the Poincaré plot along the line of identity (SD2) result from the interactions between SNS and PNS impulses [23,24]. A decrease in these variables indicates a reduced role of the PNS component [18,26,27,28,29]. Additionally, changes in the LF/HF and SD2/SD1 ratios are utilised to assess sympathovagal balance, with increases in these metrics suggesting predominant SNS activity [8,30,31]. While HRV decreases in horses during exercise, indicating a shift towards SNS activity [6,18,26], there is a subsequent shift towards increased PNS activity following aerobic training [10,32,33]. Recently, resting HR and autonomic regulation appeared to improve in geriatric horses undertaking a structured training programme [10]. HRV metrics have also been proposed as overtraining or fitness indicators in horses following typical exercise programmes [34,35]. This insight may help optimise exercise programmes to minimise the adverse effects of training on horses.

Numerous studies have shown that appropriate exercise programmes can improve resting HR and HRV in horses [10,33,36]. However, resting HRV may not effectively reflect the impact of long-term exercise programmes, as it can remain unchanged due to the complete activation of the vagal component at rest in horses [8,10]. Instead, changes in HRV related to the exercise programme may be better assessed by measuring HRV during the exercise sessions themselves. This study aimed to investigate how HR and HRV adapt during exercise sessions and recovery periods in untrained adult horses (UAHs) and untrained geriatric horses (UGHs) over a 12-week structured exercise programme. We hypothesise that HR and HRV will exhibit different patterns of adaptation during exercise and recovery in UAHs and UGHs participating in the same structured programme.

## 2. Materials and Methods

### 2.1. Horses

This study involved 18 mixed-breed UAHs and UGHs. Nine UAHs (four geldings, four mares, and one stallion, aged 8.8 ± 2.0 years, weighing 306.4 ± 75.8 kg) were selected from the teaching farms of the Laboratory Animal Centre, Kasetsart University, Nakorn Pathom, Thailand (14°00′56.0″ N 99°57′43.9″ E). Nine UGHs (five geldings and four mares, aged 20.8 ± 3.9 years, weighing 371.1 ± 51.1 kg) were selected from the Horse Lover’s Club, Pathum Thani, Thailand (13°59′38.5″ N 100°40′52.9″ E). The UAHs involved in this study had never undergone training for competitive sports or other human activities, and the UGHs had not received training since their retirement from competition at around 15 years of age. Both UAHs and UGHs were housed in individual stalls measuring 4 × 4 m and were given access to paddocks for 2–4 h daily. Their diet consisted of 2 kg of commercial pellets and 60–100 g of trace minerals, administered three times daily, along with free access to hay from hanging pergolas and tap water in their stalls. The inclusion criteria were (1) not receiving any medical or surgical treatments 30 days before or during the 12-week study period and (2) successfully passing physical, haematological, and lameness examinations before the study began. Horses were excluded if they had health issues that could affect their welfare during the experiment. Notably, two UGHs were withdrawn from this study before data collection at week 8 due to irregular gait. Therefore, data for the UGH group were obtained from seven horses from weeks 8 to 12.

### 2.2. Experimental Protocol

A 12-week structured exercise programme was conducted from January to March 2023, during which the horses were stabled. Throughout this period, relative humidity ranged from 57% to 68%, and air temperatures fluctuated between 21 °C and 31 °C. The study involved two groups of horses of different ages—UAHs (*n* = 9) and UGHs (*n* = 9)—who were assigned to perform round-pen lunges as part of a 12-week structured exercise programme (Table 1). The horses participated in 54 min exercise sessions in a 20 m-diameter round pen with a sand surface every other morning from 6:00 a.m. to 9:00 a.m., 3 to 4 times a week. The same lungers supervised the sessions for the entire 12 weeks. If any horse showed signs of exhaustion, fatigue, or lameness—indicators of compromised welfare—the exercise session would be stopped. On non-exercise days, the horses spent 2 to 4 h in the paddock.

### 2.3. Data Collection

The RR intervals were recorded biweekly to calculate the HRV variables, which offer valuable insights into the autonomic regulation of the horses. The horses were outfitted with a specialised HR monitoring device (Polar Electro Oy, Oulu, Finland). This device is easy to use and has undergone validation for accurately measuring HR and HRV in horses in various research settings [37,38,39,40]. In order to maximise the effectiveness of signal transmission, the Polar Equine Belt for Riding was carefully moistened before use. Ultrasound gel was also applied to the belt’s electrode to enhance the quality of the captured data. The Polar H10 HR sensor was securely attached to the dampened belt and positioned on the horse’s chest, with the sensor pocket placed on the middle-left side for optimal detection. It was wirelessly linked to a Polar Vantage V3 sports watch, which was configured to record the RR interval data for later analysis. Data were collected meticulously during the 54 min exercise bout, extending to a 120 min recovery period at precisely defined time points between 6:00 a.m. and 9:00 a.m. Consistency across bouts was maintained by ensuring that the same group of horses was recorded in the same order each time. These horses adhered to a carefully planned exercise schedule that was executed every other day, allowing for a systematic approach to monitoring their physiological responses.

The sports watch was then connected to the Polar FlowSync program (https://flow.polar.com/, accessed on 25 July 2023) to upload the RR interval data and export it as CSV files. These files were utilised to compute HRV variables using the Kubios HRV Scientific software version 4.1.2.1 (https://www.kubios.com/hrv-premium/; Kubios Oy, Kuopio, Finland; accessed on 23 December 2023), and the results were reported as MAT files. In order to ensure the accuracy of the aligned RR intervals, the automatic artefact correction feature was configured to exclude any missing, extra, or misaligned beat detections, as well as ectopic beats, including premature ventricular contractions or other arrhythmias in the RR interval time series. The automatic noise detection setting was adjusted to medium to filter out noise segments that could distort several consecutive beats, which could impact the accuracy of the HRV analysis. The smoothness priors method was also employed to eliminate nonstationarities in the RR interval time series. The cutoff frequency for trend removal was set at 0.035 Hz, as specified in the user guidelines (https://www.kubios.com/downloads/Kubios_HRV_Users_Guide.pdf, accessed on 23 December 2023). The HRV variables were expressed using the following three different domain methods and two indices:-Time domain method: Mean HR, mean beat-to-beat (RR) interval, SDNN, SD of the means of RR intervals in 5 min segments (SDANN), mean of the SDs of RR intervals in 5 min segments (SDNNI), RMSSD, and stress index.-Frequency domain method: LF band (0.01–0.07 Hz), HF band (0.07–0.6 Hz), and LF/HF ratio.-Nonlinear domain method: SD1, SD2, and SD2/SD1 ratio.-Autonomic nervous system indices: PNS and SNS indices.

The data are reported for 30 min before exercise; during the exercise bout consisting of exercise sessions 1 (Ex-1), 2 (Ex-2) and 3 (Ex-3); and for the 120 min recovery period, divided into 30 min intervals, at 2-week intervals for 12 consecutive weeks.

### 2.4. Statistical Analysis

The HR and HRV data were statistically analysed using GraphPad Prism (version 10.4.1; GraphPad Software Inc., San Diego, CA, USA). Due to missing HRV data, the mixed-effects model (restricted maximum likelihood: REML) with Greenhouse–Geisser correction was used to test the effects of independent group (elapsed time points within a training session of 120 min), independent time (periods of 12-week training programme) and the group-by-time interaction on HR and HRV changes in the UAH and UGH groups. Tukey’s multiple comparisons test was later implemented to estimate differences within and between the groups at specific times. The results are expressed as the mean ± SD. Statistical significance was set at *p* < 0.05.

## 3. Results

### 3.1. Mean HR

Changes in the HR in UAHs and UGHs in response to the 12-week structured exercise programme are shown in Table 2. The group-by-time interaction, group, and time effects were significant for changes in HR in UAHs (interaction: F_(42, 390)_ = 2.567, *p* < 0.0001; group: F_(6, 56)_ = 4.734, *p* = 0.0006; time: F_(7, 390)_ = 963.0, *p* < 0.0001). However, only the group-by-time interaction and time effects were significant for HR changes in UGHs (interaction: F_(42, 321)_ = 1.455, *p* = 0.0400; time: F_(7, 321)_ = 750.0, *p* < 0.0001).

In UAHs, the mean HR increased during the exercise sessions (Ex-1, Ex-2, and Ex-3), extending to the first 30 min of the recovery period after the initial exercise bout (0 weeks; *p* < 0.001–0.0001). The increase in mean HR was limited to just the exercise session at weeks 2–12 (*p* < 0.0001 for all pairwise comparisons). Additionally, the mean HR was lower in weeks 10 and 12 in Ex-1 (*p* < 0.05), weeks 2–12 in Ex-2 (*p* < 0.01–0.0001), and weeks 2–12 in Ex-3 (*p* < 0.05–0.0001) compared to week 0 (the initial exercise session). Conversely, in UGHs, the mean HR increased during the exercise session, extending to the first 60 min of the recovery period after the initial exercise bout (week 0; *p* < 0.0001 for all pairwise comparisons) and persisted at week 2 (*p* < 0.05–0.0001). The increase in mean HR was limited to the exercise sessions and the first 30 min of the recovery period at weeks 4–12 (*p* < 0.05–0.0001). Additionally, the mean HR was lower at weeks 10 and 12 in Ex-1 (*p* < 0.05) and weeks 6–12 in Ex-2 (*p* < 0.05–0.001) compared to week 0.

### 3.2. Mean RR Interval, SDNN, RMSSD, LF, HF, LF/HF Ratio, and SD2/SD1 Ratio

The changes in mean RR, SDNN, RMSSD, LF, HF, LF/HF ratio, and SD2/SD1 ratio in response to the 12-week structured exercise programme in UAHs and UGHs are shown in Table 3. Only time had a significant effect on the changes in various HRV variables during the exercise sessions and recovery period in UAHs and UGHs: mean RR (UAH: F_(2.492, 138.8)_ = 496.4, *p* < 0.0001; UGH: F_(3.061, 140.4)_ = 462.5, *p* < 0.0001), SDNN (UAH: F_(2.235, 124.5)_ = 52.52, *p* < 0.0001; UGH: F_(7, 321)_ = 87.97, *p* < 0.0001), RMSSD (UAH: F_(2.159, 120.3)_ = 32.58, *p* < 0.0001; UGH: F_(1.914, 87.75)_ = 49.17, *p* < 0.0001), LF band (UAH: F_(5.030, 280.2)_ = 24.40, *p* < 0.0001; UGH: F_(4.664, 213.9)_ = 49.15, *p* < 0.0001), HF band (UAH: F_(5.014, 279.4)_ = 24.79, *p* < 0.0001; UGH: F_(4.679, 214.6)_ = 49.13, *p* < 0.0001), LF/HF ratio (UAH: F_(4.200, 234.0)_ = 18.98, *p* < 0.0001; UGH: F_(2.968, 136.1)_ = 46.55, *p* < 0.0001), and SD2/SD1 ratio (UAH: F_(3.281, 182.8)_ = 200.3, *p* < 0.0001; UGH: F_(2.986, 136.9)_ = 108.8, *p* < 0.0001).

The mean RR interval decreased during exercise sessions in both UAHs and UGHs compared to the pre-exercise (baselines) value (*p* < 0.0001 for all pairwise comparisons). However, the decrease in the mean RR interval was limited to the first 60 min of the recovery period in the UAHs (*p* < 0.05–0.0001) but extended to the full 120 min recovery period in the UGHs (*p* < 0.0001 for all pairwise comparisons). SDNN and RMSSD both decreased during the exercise sessions, extending to the first 30 min of the recovery period in UAHs (SDNN: *p* < 0.001–0.0001, RMSSD: *p* < 0.01–0.0001), but the first 60 min in UGHs (SDNN: *p* < 0.0001, RMSSD: *p* < 0.05–0.0001). The HF band decreased, corresponding to an increase in the LF band and LF/HF ratio during Ex-1 and Ex-2 in UAHs (HF and LF: *p* < 0.0001, LF/HF ratio: *p* < 0.001–0.0001 for all pairwise comparisons in both periods). In contrast, the changes in three variables extended to Ex-3 in UGHs (*p* < 0.0001 for all pairwise comparisons). The SD2/SD1 ratio increased during exercise sessions, extending to the first 30 min of the recovery period in UAHs (*p* < 0.05–0.0001). The SD2/SD1 ratio also increased during exercise sessions but extended to the first 90 min of the recovery period in UGHs (*p* < 0.05–0.0001).

### 3.3. SDANN, SDNNI, SD1, SD2, and PNS Index

Changes in the SDANN, SDNNI, SD1, SD2, and PNS index in response to the 12-week structured exercise programme in UAHs and UGHs are shown in Table 4, Table 5 and Table 6. In UAHs, time had a significant effect on the changes in SDANN (F_(3.597, 195.8)_ = 25.14, *p* < 0.0001), SDNNI (F_(3.607, 167.5)_ = 75.30, *p* < 0.0001), SD1 (F_(2.158, 120.2)_ = 32.53, *p* < 0.0001), SD2 (F_(2.346, 130.7)_ = 77.98, *p* < 0.0001), and PNS index (F_(2.152, 119.9)_ = 179.2, *p* < 0.0001). However, in UGHs, the group-by-time interaction, group, and time had a significant effect on changes in SD1 (interaction: F_(42, 321)_ = 1.573, *p* = 0.0169; group: F_(6, 46)_ = 3.515, *p* = 0.0060; time: F_(1.997, 91.56)_ = 57.12, *p* < 0.0001). In addition, the group-by-time interaction and time had a significant effect on changes in SDANN (interaction: F_(42, 282)_ = 1.553, *p* = 0.0208; time: F_(7, 282)_ = 32.28, *p* < 0.0001), SDNNI (interaction: F_(42, 282)_ = 1.433, *p* = 0.0485; time: F_(3.654, 147.2)_ = 47.17, *p* < 0.0001), SD2 (interaction: F_(42, 321)_ = 1.425, *p* = 0.0493; time: F_(3.073, 140.9)_ = 119.4, *p* < 0.0001), and PNS index (interaction: F_(42, 321)_ = 1.438, *p* = 0.0449; time: F_(2.336, 107.1)_ = 269.0, *p* < 0.0001).

In UAHs, the SDANN increased in Ex-1 (*p* < 0.05) but decreased in Ex-2 (*p* < 0.05) and then returned to baseline in Ex-3 and throughout the 120 min recovery period. The SDNNI decreased from Ex-1 to the 120 min recovery period (*p* < 0.0001 for all pairwise comparisons). The SD1 and PNS index decreased in Ex-1, extending to the first 30 min of the recovery period (*p* < 0.01–0.0001), while SD2 decreased in Ex-1, extending to the first 60 min of the recovery period (*p* < 0.01–0.0001; Table 4).

Unlike in UAHs, the SDANN, SDNNI SD1, SD2, and PNS indexes fluctuated over the 12 weeks of the exercise programme in UGHs. The SDANN decreased during Ex-2 and Ex-3 in weeks 0 and 2 (*p* < 0.05–0.01). It occasionally increased in Ex-1 in week 6 (*p* < 0.01) and in the first 30 min of the recovery period in weeks 6–12 (*p* < 0.01–0.0001). The SDANN was decreased pre-exercise in weeks 2–10 (*p* < 0.05 for all pairwise comparisons) but increased in the first 30 min of the recovery period in weeks 6–12 (*p* < 0.01 for all comparison pairs). The SDNNI was decreased from Ex-1 to the first 30 min of the 12 min recovery period in week 2 (*p* < 0.05–0.01). It also decreased in Ex-3 in week 4 (*p* < 0.05) and Ex-1–3 in week 6 (*p* < 0.05–0.01). The SDNNI was higher during the first 30 min of the recovery period in week 12 compared to week 0 (*p* < 0.05; Table 5).

SD1 was decreased in Ex-1, extending to the first 30 min of the recovery period in weeks 0–6 (*p* < 0.05–0.0001). However, the decrease in SD1 was limited to Ex-1–3 in weeks 8–12 (*p* < 0.05–0.0001). Compared to week 0, SD1 increased from Ex-2 to the full 120 min recovery period in week 2 (*p* < 0.05–0.01). SD2 decreased in Ex-1, extending to the first 30 min of the recovery period in weeks 0–2 (*p* < 0.05–0.001). However, the decrease in SD2 was limited to Ex-1–3 in weeks 4–12 (*p* < 0.05–0.001). The PNS index decreased in Ex-1, extending to the first 60 min of the recovery period in weeks 0, 2, 8, and 10 (*p* < 0.05–0.0001). However, the decrease in the PNS index was limited to Ex-1 to the first 30 min of the recovery period in weeks 4, 6, and 12 (*p* < 0.05–0.0001; Table 6).

### 3.4. SNS Index

The group-by-time interaction, group, and time had a significant effect on changes in the SNS index in UAHs (interaction: F_(42, 390)_ = 4.076, *p* < 0.0001; group: F_(6, 56)_ = 4.916, *p* = 0.0004; time: F_(2.493, 138.9)_ = 720.9, *p* < 0.0001). In contrast, only group and time had a significant effect on changes in the SNS index in UGHs (group: F_(6, 46)_ = 2.347, *p* = 0.0463; time: F_(2.333, 107.0)_ = 337.6, *p* <0.0001; Table 7).

In UAHs, the SNS index increased from Ex-1 to the first 30 min of the recovery period in week 0 (*p* < 0.05–0.001). However, the increase in the SNS index was limited to Ex-1 and Ex-3 in weeks 2–12 (*p* < 0.01–0.001). In contrast, the SNS index decreased in Ex-2 in weeks 2–12 (*p* < 0.05–0.0001). In UGHs, the SNS index was increased from Ex-1 to the first 90 min of the recovery period (*p* < 0.05–0.001). However, the increase in the SNS index was limited to Ex-1 to the first 30 min of the recovery period in weeks 2–10 (*p* < 0.05–0.0001) and then to Ex-1 to Ex-3 in week 12 (*p* < 0.05–0.01).

### 3.5. Stress Index

The group-by-time interaction, group, and time had a significant effect on changes in the stress index in UAHs (interaction: F_(42, 390)_ = 2.565, *p* < 0.0001; group: F_(6, 56)_ = 2.525, *p* = 0.0311; time: F_(2.372, 132.2)_ = 300.2, *p* < 0.0001) and UGHs (interaction: F_(42, 321)_ = 1.486, *p* = 0.0322; group: F_(6, 46)_ = 2.923, *p* = 0.0169; time: F_(2.156, 98.87)_ = 235.0, *p* < 0.0001; Table 8).

In UAHs, the stress index was increased from Ex-1 to Ex-3 in weeks 0–12 (*p* < 0.05–0.0001). The stress index was lower in Ex-2 in weeks 10–12 compared to week 0 (*p* < 0.05–0.01). In contrast, in UGHs, the stress index was increased from Ex-1 to the first 90 min of the recovery period in week 0 (*p* < 0.05–0.0001). The increase in the stress index was then limited to Ex-1 to the first 30 min of the recovery period in weeks 2–10 and then to Ex-1 to Ex-3 in week 12 (*p* < 0.05–0.001).

## 4. Discussion

Our study examined changes in HR and autonomic regulation in UAHs and UGHs during exercise sessions and recovery periods within a 12-week structured exercise programme. Our key findings are as follows. Firstly, the duration of elevated mean HR, SNS index, and stress index during the 54 min exercise bout gradually decreased over the 12 weeks for both UAHs and UGHs. Secondly, the decreases in these three variables occurred faster in the UAHs than in the UGHs. Thirdly, while mean HR decreased in both UAHs and UGHs during the exercise bout (Ex-1 to Ex-3), the SNS and stress indices only decreased in UAHs over the 12 weeks. Fourthly, the HRV measures mean RR interval, SDNN, RMSSD, LF, HF, LF/HF ratio, and SD2/SD1 ratio returned to baseline faster in UAHs than in UGHs during the 54 min exercise bout. Fifthly, while the SDANN, SDNNI, SD1, SD2, and PNS indexes did not change significantly between exercise periods in UAHs, they fluctuated over the 12 weeks in UGHs. These results indicate that the 12-week structured exercise programme positively influenced HR and HRV adaptation in both UAHs and UGHs, with each exhibiting distinct dynamic adaptations.

Research has shown that resting HR decreases in horses due to cardiovascular adaptation following proper exercise training [7,10,33]. However, in our study, the resting HRs measured before the start of exercise bouts every 2 weeks did not differ significantly. Nevertheless, resting HR appeared to decrease over the 12-week exercise programme in both UAHs and UGHs. This lack of a significant change may be attributed to individual variations in behavioural and HRV parameters reported in horses [41,42,43]. Such variations could also underlie the observed inconsistencies within each group, particularly since they are untrained horses, resulting in nonsignificant changes in resting HR over the 12 weeks.

In contrast, the mean HR exhibited a different trend during exercise sessions. In the initial exercise bout (week 0), the mean HR was 114–167 beats per minute among UAHs compared to 123–161 beats per minute among UGHs during the exercise sessions (Ex-1, Ex-2, and Ex-3). At the end of the 12-week structured exercise programme, the mean HR had decreased to 96–126 beats per minute among UAHs and 98–130 beats per minute among UGHs. Therefore, the 12-week structured exercise programme appeared to improve HRs.

The increase in HR during exercise is primarily due to SNS activity [43,44]. It has been suggested that SNS activity might partially decrease after exercise training, contributing to a consistent reduction in mean HR during the exercise sessions while using the same training structure. As expected, UGHs exhibited a slower return to baseline HR than UAHs after the 12-week structured exercise programme. Since research indicates that older horses show a reduced physiological response [45], a later reduction in the duration of increased HR reflects a slower adaptation to exercise training among UGHs.

In addition to mean HR, HRV variables were measured to assess the impact of long-term exercise programmes. Research has shown that decreased HRV in horses indicates stress during exercise [6,8,18,21,26,44]. Several studies have also utilised HRV changes to evaluate the effects of training in horses [28,32,33]. In our study, various HRV metrics, including mean RR interval, SDNN, RMSSD, LF band, and HF band, as well as the sympathovagal balance indicated by the LF/HF and SD2/SD1 ratios, did not change significantly across the 12-week structured exercise programme for both UAHs and UGHs. However, these HRV variables returned to baseline slower in the UGHs than in the UAHs during the 54 min exercise bouts. This finding suggests that UGHs experienced a slower recovery than UAHs after the structured exercise bouts.

The SDANN, SDNNI, SD1, SD2, and the PNS index did not change significantly over the 12-week structured exercise programme in UAHs. However, some HRV metrics did change over the 12 weeks in UGHs. Changes were observed in SDANN and SDNN. Since these metrics were assessed in 5 min segments during the 54 min exercise bouts, it is plausible that transient fluctuations in RR intervals occurred throughout the bouts. These fluctuations may have contributed to the significant variations observed in the 5 min segments in UGHs. Such transient changes could explain the overall increase in SDNN across the exercise sessions, consistent with previous studies [29,46].

In addition to SDANN and SDNNI, SD1 and SD2 also varied in UGHs. The geometric measures of SD1 and SD2 can define the complexity of cardiac rhythm, which may not be captured by other HRV measures [22,24]. Therefore, the unchanged values of several HRV variables computed using time and frequency domain methods in our study may not fully reflect the underlying dynamics. Consequently, the variations observed in SD1 and SD2 in UGHs but not UAHs imply greater complexity and variability in RR intervals among UGHs than in UAHs.

Moreover, the PNS index changed over the 12-week structured exercise programme in UGHs. In UAHs, the PNS index decreased during exercise and the first 30 min of the recovery period throughout the 12 weeks. However, in UGHs, the PNS index was only decreased during the first 30 min of the recovery period in weeks 4, 6, and 12. Since PNS saturation has been observed in young horses undergoing exercise training [33], PNS tone may not have been fully activated in the UGHs, leading to these variations during training. Nonetheless, the changes in the PNS index in the UGHs indicated improved PNS activity after the 12-week structured exercise programme.

While the stress and SNS indices varied in both UAHs and UGHs, the extent of this variation differed between them. Although the stress and SNS indices were only increased in the three exercise sessions in weeks 0 and 2 in UAHs, they were increased up to week 12 in the UGHs, suggesting that the UGHs adapted to a structured exercise programme more slowly than the UAHs. Notably, while the SNS index varied in UAHs, both the SNS and PNS indices varied in the UGHs. Since HR is influenced by interactions between the SNS and PNS components [22,23,24], it can be primarily modulated by the SNS in response to appropriate exercise training in horses [8,36]. Our findings are consistent with previous studies indicating that variations in the SNS index but not the PNS index may indicate full activation of the PNS component, leading the SNS component to significantly influence (decrease) the HR in adult horses. In contrast, the simultaneous changes in both the PNS and SNS indices in UGHs may reflect the combined effects of both the PNS and SNS in lowering HR. Overall, our findings reinforce the notion that more pronounced physiological responses are linked to slower adaptation to structured exercise programmes in UGHs compared to UAHs.

Our study had some limitations. Firstly, it could not compute average speed and distance, as the measuring device could not be used on the horses during the exercise sessions, which involved round-pen lunging without lunging aids or riders. Therefore, our study does not provide information on the average speed and distance of the horses during the exercise sessions. Secondly, the considerable variation among the horses could lead to significant differences in HRV metrics within the same group. Therefore, the HRV measurements reported in this study should be interpreted cautiously.

## 5. Conclusions

A 12-week structured exercise programme positively impacted HR and HRV metrics during both the exercise sessions and recovery periods in UAHs and UGHs. However, the extent of changes in these parameters differed between UAHs and UGHs, with UGHs showing slower adaptation to the exercise programme than UAHs. Therefore, changes in HR and HRV during exercise and recovery could serve as valuable indicators for assessing training effects and distinguishing adaptation levels among horses with varying conditioning that follow similar exercise programmes.

## Figures and Tables

**Table 1 animals-15-01122-t001:** The 54 min structured exercise bout.

Order	Gait	Minutes	Exercise Session
1	Walking	5	Ex-1
2	Trotting	10	
3	Walking	1	
4	Trotting	10	
5	Walking	1	
6	Cantering	5	Ex-2
7	Walking	1	
8	Cantering	5	
9	Walking	1	
10	Trotting	10	Ex-3
11	Walking	5	

**Table 2 animals-15-01122-t002:** The heart rate (HR; mean ± standard deviation [SD]) in untrained adult horses (UAHs) and geriatric horses (UGHs) before, during, and after exercise bouts at 2-week intervals over the 12-week structured exercise programme.

Groups	Stage	Pre- Exercise	Exercise			Post-Exercise			
			Ex-1	Ex-2	Ex-3	30 min	60 min	90 min	120 min
UAH ***^,#,¥^**	0 wks	41.2 ± 6.2	113.9 ± 18.5 **^α^**	166.9 ± 12.6 **^α^**	122.8 ± 14.1 **^α^**	61.0 ± 60.4 **^α^**	51.0 ± 11.6	49.1 ± 7.7	48.7 ± 8.6
	2 wks	43.4 ± 12.6	107.0 ± 13.4 **^α^**	138.3 ± 17.5 **^α,x^**	106.0 ± 25.2 **^α,x^**	51.4 ± 7.1	44.0 ± 8.4	41.8 ± 6.1	40.7 ± 4.6
	4 wks	46.8 ± 14.7	107.3 ± 15.4 **^α^**	145.4 ± 26.8 **^α,x^**	112.1 ± 15.5 **^α,x^**	50.1 ± 8.9	42.2 ± 8.4	40.0 ± 8.1	42.4 ± 9.4
	6 wks	40.6 ± 6.4	103.0 ± 11.8 **^α^**	133.3 ± 18.6 **^α,x^**	104.4 ± 18.3 **^α,x^**	53.4 ± 9.6	46.8 ± 9.1	42.8 ± 6.9	42.1 ± 5.6
	8 wks	38.8 ± 5.3	102.6 ± 13.0 **^α^**	132.4 ± 14.1 **^α^**^,**x**^	103.9 ± 14.7 **^α^**^,**x**^	52.1 ± 7.3	45.0 ± 6.4	43.9 ± 7.2	42.4 ± 7.0
	10 wks	37.4 ± 4.2	95.4 ± 11.8 **^α,x^**	125.4 ± 11.3 **^α,x^**	94.9 ± 9.7 **^α,x^**	48.4 ± 5.5	45.1 ± 6.6	42.0 ± 5.8	39.2 ± 3.9
	12 wks	39.9 ± 3.5	96.4 ± 6.9 **^α,x^**	126.1 ± 13.7 **^α,x^**	97.1 ± 9.9 **^α,x^**	49.8 ± 7.7	47.0 ± 10.6	46.1 ± 8.3	45.9 ± 8.9
UGH ***^,¥^**	0 wks	42.8 ± 7.9	122.9 ± 19.9 **^α^**	160.6 ± 24.2 **^α^**	129.3 ± 29.1 **^α^**	76.6 ± 16.8 **^α^**	66.1 ± 14.7 **^α^**	56.1 ± 13.9	51.1 ± 11.4
	2 wks	40.3 ± 8.7	119.4 ± 29.6 **^α^**	150.7 ± 28.8 **^α^**	125.9 ± 24.1 **^α^**	65.1 ± 17.4 **^α^**	55.2 ± 18.8 **^α^**	50.0 ± 15.1	46.8 ± 9.1
	4 wks	41.7 ± 9.9	107.2 ± 12.3 **^α^**	143.0 ± 22.7 **^α^**	116.3 ± 15.0 **^α^**	60.4 ± 10.9 **^α^**	49.3 ± 8.1	43.0 ± 9.5	40.4 ± 5.8
	6 wks	39.4 ± 7.6	101.0 ± 14.7 **^α^**	132.1 ± 21.7 **^α,x^**	113.6 ± 15.2 **^α^**	57.6 ± 9.0 **^α^**	54.8 ± 12.9	50.8 ± 9.4	47.6 ± 5.4
	8 wks	40.0 ± 7.9	105.5 ± 10.9 **^α^**	133.0 ± 19.9 **^α,x^**	113.3 ± 14.2 **^α^**	56.7 ± 10.7 **^α^**	45.5 ± 9.5	47.0 ± 12.2	42.7 ± 7.7
	10 wks	37.2 ± 7.5	95.7 ± 11.2 **^α,x^**	125.0 ± 13.6 **^α,x^**	110.0 ± 29.1 **^α^**	57.7 ± 15.0 **^α^**	46.3 ± 11.8	44.0 ± 9.9	42.3 ± 7.7
	12 wks	37.0 ± 6.4	98.2 ± 11.9 **^α,x^**	130.7 ± 16.7 **^α,x^**	110.0 ± 20.1 **^α^**	55.0 ± 11.8 **^α^**	47.3 ± 11.9	45.7 ± 8.5	41.7 ± 7.1

***^,#,¥^**: significant group-by-time interaction, independent group, and independent time effects; **^α^**: within-group differences compared to pre-exercise values; **^x^**: between-group differences at each exercise session compared to the value at week 0; HR: heart rate; UAH: untrained adult horse; UGH: untrained geriatric horse; wks, week.

**Table 3 animals-15-01122-t003:** Changes in heart rate variables (mean ± SD) in UAHs and UGHs before, during, and after exercise bouts.

Variables	Group	Pre- Exercise	Exercise			Post-Exercise			
			Ex-1	Ex-2	Ex-3	30 min	60 min	90 min	120 min
Mean RR interval (ms) **^¥^**	UAH	1506.1 ± 81.7	589.1 ± 34.6 **^α^**	443.8 ± 41.8 **^α^**	585.6 ± 50.8 **^α^**	1172.6 ± 86.5 **^α^**	1353.5 ± 71.5 **^α^**	1413.6 ± 97.9	1434.1 ± 92.6
	UGH	1560.5 ± 80.2	568.6 ± 44.3 **^α^**	440.4 ± 34.4 **^α^**	528.2 ± 31.8 **^α^**	1027.1 ± 106.1 **^α^**	1230.1 ± 144.7 **^α^**	1315.5 ± 111.5 **^α^**	1387.1 ± 109.4 **^α^**
SDNN (ms) **^¥^**	UAH	85.6 ± 18.3	26.7 ± 2.3 **^α^**	13.9 ± 2.6 **^α^**	15.1 ± 2.1 **^α^**	53.3 ± 4.8 **^α^**	65.5 ± 7.2	77.2 ± 9.9	77.6 ± 13.8
	UGH	82.1 ± 8.5	22.8 ± 4.2 **^α^**	10.9 ± 2.1 **^α^**	11.7 ± 3.1 **^α^**	45.8 ± 6.6 **^α^**	60.6 ± 13.5 **^α^**	70.2 ± 15.6	76.3 ± 13.8
RMSSD (ms) **^¥^**	UAH	89.4 ± 28.8	14.0 ± 1.4 **^α^**	5.7 ± 0.9 **^α^**	7.9 ± 1.6 **^α^**	52.2 ± 7.1 **^α^**	71.0 ± 12.4	86.1 ± 15.7	82.2 ± 20.8
	UGH	89.0 ± 12.3	12.6 ± 2.8 **^α^**	4.7 ± 1.0 **^α^**	6.5 ± 2.0 **^α^**	37.4 ± 7.2 **^α^**	55.8 ± 20.0 **^α^**	70.6 ± 25.8	79.4 ± 22.6
LF band (%) **^¥^**	UAH	50.3 ± 5.7	64.4 ± 2.9 **^α^**	64.5 ± 4.5 **^α^**	53.6 ± 2.8	46.7 ± 3.4	47.1 ± 2.1	49.3 ± 1.9	50.8 ± 3.3
	UGH	46.2 ± 2.7	69.7 ± 2.6 **^α^**	69.7 ± 4.9 **^α^**	60.5 ± 3.5 **^α^**	46.3 ± 3.9	46.1 ± 3.6	46.3 ± 3.4	46.5 ± 4.9
HF band (%) **^¥^**	UAH	49.7 ± 5.7	35.6 ± 2.9 **^α^**	35.2 ± 4.5 **^α^**	46.3 ± 2.7	53.3 ± 3.4	52.9 ± 2.1	50.7 ± 1.9	49.2 ± 3.3
	UGH	53.7 ± 2.6	30.2 ± 2.6 **^α^**	30.2 ± 4.9 **^α^**	39.5 ± 3.5 **^α^**	53.7 ± 3.8	53.9 ± 3.6	53.7 ± 3.4	53.5 ± 4.9
LF/HF ratio **^¥^**	UAH	1.3 ± 0.3	2.3 ± 0.3 **^α^**	2.3 ± 0.5 **^α^**	1.4 ± 0.1	1.0 ± 0.2	1.1 ± 0.2	1.1 ± 0.1	1.3 ± 0.3
	UGH	1.0 ± 0.1	2.6 ± 0.3 **^α^**	2.9 ± 0.5 **^α^**	1.9 ± 0.4 **^α^**	1.0 ± 0.1	1.0 ± 0.1	1.0 ± 0.1	1.0 ± 0.2
SD2/SD1 ratio **^¥^**	UAH	1.8 ± 0.2	3.9 ± 0.2 **^α^**	5.0 ± 0.4 **^α^**	3.9 ± 0.3 **^α^**	2.1 ± 0.1 **^α^**	1.9 ± 0.1	1.9 ± 0.1	1.9 ± 0.1
	UGH	1.7 ± 0.1	3.9 ± 0.3 **^α^**	4.7 ± 0.3 **^α^**	3.8 ± 0.4 **^α^**	2.4 ± 0.3 **^α^**	2.2 ± 0.3 **^α^**	2.0 ± 0.2 **^α^**	1.9 ± 0.2

**^¥^**: significant time effect; **^α^**: within-group differences compared to pre-exercise value; RR: beat-to-beat interval; SDNN: standard deviation of normal-to-normal RR intervals; RMSSD: root mean square of differences between successive RR intervals; LF: low-frequency band; HF: high-frequency band; SD1: standard deviation of the Poincaré plot perpendicular to the line of identity; SD2: standard deviation of the Poincaré plot along the line of identity.

**Table 4 animals-15-01122-t004:** Changes in SDANN, SDNNI, SD1, SD2, and PNS index in UAHs before, during, and after the exercise bouts.

Variables	Pre- Exercise	Exercise			Post-Exercise			
		Ex-1	Ex-2	Ex-3	30 min	60 min	90 min	120 min
SDANN (ms) **^¥^**	94.7 ± 48.5	121.5 ± 10.2 **^α^**	14.9 ± 2.1 **^α^**	25.9 ± 3.5	141.0 ± 16.7	74.1 ± 18.6	79.7 ± 21.8	68.0 ± 19.9
SDNNI (ms) **^¥^**	179.7 ± 36.6	65.5 ± 2.4 **^α^**	56.7 ± 8.4 **^α^**	42.1 ± 8.0 **^α^**	100.3 ± 10.3 **^α^**	116.2 ± 5.5 **^α^**	133.5 ± 20.5 **^α^**	132.1 ± 17.3 **^α^**
SD1 (ms) **^¥^**	63.4 ± 20.4	9.9 ± 1.0 **^α^**	4.0 ± 0.7 **^α^**	5.6 ± 1.1 **^α^**	37.0 ± 5.0 **^α^**	50.2 ± 8.8	60.9 ± 11.1	58.2 ± 14.8
SD2 (ms) **^¥^**	102.3 ± 16.1	36.6 ± 2.8 **^α^**	19.5 ± 3.6 **^α^**	20.8 ± 2.8 **^α^**	64.9 ± 5.1 **^α^**	76.4 ± 6.1 **^α^**	88.0 ± 9.6	91.6 ± 12.5
PNS index **^¥^**	3.9 ± 1.0	−2.5 ± 0.2 **^α^**	−3.7 ± 0.3 **^α^**	−2.7 ± 0.4 **^α^**	1.4 ± 0.5 **^α^**	2.7 ± 0.6	3.4 ± 0.8	3.4 ± 0.8

**^¥^**: significant time effects; **^α^**: within-group differences compared to pre-exercise value; SDANN: standard deviation of the means of RR intervals in 5 min segments; SDNNI: mean of the standard deviations of RR intervals in 5 min segments; SD1: standard deviation of the Poincaré plot perpendicular to the line of identity; SD2: standard deviation of the Poincaré plot along the line of identity; PNS: parasympathetic nervous system.

**Table 5 animals-15-01122-t005:** Changes in SDANN and SDNNI (mean ± SD) in UGHs before, during, and after exercise bouts at 2-week intervals during the 12-week structured exercise programme.

Variables	Stage	Pre- Exercise	Exercise			Post-Exercise			
			Ex-1	Ex-2	Ex-3	30 min	60 min	90 min	120 min
SDANN ***^,¥^**	0 wks	132.0 ± 122.5	128.4 ± 37.6	12.8 ± 9.9 **^α^**	25.5 ± 36.0 **^α^**	87.3 ± 42.3	60.5 ± 48.4	88.7 ± 46.6	66.6 ± 40.4
	2 wks	111.8 ± 135.6 **^x^**	104.2 ± 61.8	19.1 ± 11.3 **^α^**	26.0 ± 33.4 **^α^**	159.7 ± 81.5	74.5 ± 62.7	80.2 ± 59.2	109.0 ± 57.0
	4 wks	34.5 ± 24.7 **^x^**	118.7 ± 31.2	7.8 ± 3.8	14.1 ± 13.8	162.3 ± 50.0	97.8 ± 64.2	79.6 ± 34.7	66.0 ± 26.1
	6 wks	42.3 ± 30.4 **^x^**	144.4 ± 88.3 **^α^**	20.9 ± 25.7	46.0 ± 36.7	189.0 ± 72.2 **^α,x^**	77.9 ± 40.3	62.9 ± 48.4	74.0 ± 55.0
	8 wks	33.2 ± 8.1 **^x^**	101.7 ± 43.4	9.9 ± 5.3	27.4 ± 14.7	201.8 ± 74.3 **^α,x^**	112.9 ± 60.1	58.3 ± 24.1	78.8 ± 40.7
	10 wks	31.6 ± 33.4 **^x^**	123.5 ± 54.3	42.3 ± 25.1	52.0 ± 39.0	148.0 ± 97.6 **^α,x^**	92.4 ± 53.2	92.9 ± 83.8	111.0 ± 88.7
	12 wks	61.8 ± 91.1	151.7 ± 27.7	17.0 ± 13.1	25.5 ± 23.4	195.3 ± 77.0 **^α,x^**	64.7 ± 37.0	72.9 ± 61.0	74.8 ± 39.2
SDNNI ***^,¥^**	0 wks	184.4 ± 116.4	68.3 ± 36.8	42.5 ± 9.4	23.8 ± 15.3	66.8 ± 35.5	82.4 ± 49.4	98.2 ± 47.4	105.9 ± 42.3
	2 wks	170.9 ± 48.9	65.8 ± 28.6 **^α^**	58.2 ± 22.6 **^α^**	29.8 ± 9.5 **^α^**	96.9 ± 29.0 **^α^**	99.5 ± 32.4	131.2 ± 63.9	152.0 ± 58.9
	4 wks	104.8 ± 30.4	58.9 ± 31.0	62.2 ± 13.8	34.0 ± 15.4 **^α^**	102.9 ± 29.9	141.4 ± 87.9	158.2 ± 84.9	127.1 ± 33.2
	6 wks	163.5 ± 55.5	83.8 ± 42.5 **^α^**	75.3 ± 37.7 **^α^**	37.3 ± 9.1 **^α^**	104.3 ± 23.5	101.0 ± 21.8	97.9 ± 25.4	126.3 ± 43.3
	8 wks	136.9 ± 40.0	62.6 ± 15.8	73.3 ± 22.8	40.4 ± 8.3	105.7 ± 18.9	119.3 ± 22.1	107.8 ± 32.7	134.9 ± 55.8
	10 wks	170.7 ± 78.7	79.4 ± 21.4	75.8 ± 37.5	51.6 ± 20.6	115.5 ± 41.0	123.6 ± 23.0	134.3 ± 46.8	131.5 ± 38.2
	12 wks	125.1 ± 69.7	69.4 ± 16.6	75.2 ± 24.4	44.8 ± 19.1	115.6 ± 13.2 **^x^**	103.9 ± 35.5	115.5 ± 25.8	117.1 ± 32.6

***^,¥^**: significant group-by-time interaction and time effects; **^α^**: within-group differences compared to pre-exercise value; **^x^**: between-group differences at each exercise session compared to the week 0 value; SDANN: standard deviation of the means of RR intervals in 5 min segments; SDNNI: mean of the standard deviations of RR intervals in 5 min segments; wks: weeks.

**Table 6 animals-15-01122-t006:** Changes in SD1, SD2, and PNS index (mean ± SD) in UGHs before, during, and after exercise bouts at 2-week intervals over the 12-week structured exercise programme.

Variables	Stage	Pre- Exercise	Exercise			Post-Exercise			
			Ex-1	Ex-2	Ex-3	30 min	60 min	90 min	120 min
SD1 (ms) ***^,¥^**	0 wks	75.78 ± 40.3	9.2 ± 6.5 **^α^**	2.7 ± 1.3 **^α^**	2.9 ± 1.1 **^α^**	17.4 ± 12.0 **^α^**	22.9 ± 16.1	35.5 ± 27.2	47.2 ± 29.6
	2 wks	98.9 ± 25.9	25.2 ± 13.4 **^α^**	12.2 ± 5.0 **^α,x^**	12.7 ± 6.3 **^α,x^**	54.4 ± 18.3 **^α,x^**	60.4 ± 19.5 **^x^**	89.8 ± 28.3 **^x^**	99.9 ± 34.1 **^x^**
	4 wks	63.4 ± 40.8	7.6 ± 3.1 **^α^**	2.9 ± 1.0 **^α^**	4.5 ± 2.9 **^α^**	28.3 ± 12.4 **^α^**	65.4 ± 79.1	87.8 ± 95.8	90.7 ± 92.8
	6 wks	57.1 ± 24.3	9.3 ± 5.0 **^α^**	3.6 ± 2.3 **^α^**	4.2 ± 1.6 **^α^**	29.6 ± 10.9 **^α^**	33.1 ± 10.2	37.2 ± 20.9	43.8 ± 17.5
	8 wks	61.6 ± 24.5	6.8 ± 3.1 **^α^**	3.6 ± 2.5 **^α^**	4.7 ± 2.9 **^α^**	26.4 ± 4.9	39.5 ± 11.4	37.4 ± 12.0	48.4 ± 14.1
	10 wks	71.7 ± 39.5	10.6 ± 5.2 **^α^**	4.6 ± 3.6 **^α^**	6.8 ± 4.0 **^α^**	31.6 ± 10.3	48.4 ± 19.7	52.6 ± 20.9	53.8 ± 20.7
	12 wks	48.6 ± 6.9	12.0 ± 4.1 **^α^**	3.5 ± 2.0 **^α^**	5.9 ± 4.6 **^α^**	30.1 ± 10.7	39.6 ± 17.8	45.8 ± 14.8	50.0 ± 15.1
SD2 (ms) ***^,¥^**	0 wks	108.1 ± 32.3	27.2 ± 11.7 **^α^**	11.2 ± 6.0 **^α^**	9.3 ± 4.8 **^α^**	43.6 ± 26.0 **^α^**	55.2 ± 32.2	61.7 ± 26.9	76.3 ± 22.6
	2 wks	98.9 ± 25.9	25.2 ± 13.4 **^α^**	12.2 ± 5.0 **^α^**	12.7 ± 6.3 **^α^**	54.4 ± 18.3 **^α^**	60.4 ± 19.5	89.8 ± 28.3	99.9 ± 34.1
	4 wks	88.2 ± 34.8	25.7 ± 7.3 **^α^**	13.5 ± 8.2 **^α^**	16.7 ± 6.5 **^α^**	56.7 ± 20.5	94.2 ± 58.3	108.8 ± 68.1	113.4 ± 60.2
	6 wks	87.0 ± 26.1	32.5 ± 11.8 **^α^**	16.3 ± 8.5 **^α^**	14.1 ± 5.9 **^α^**	62.5 ± 11.7	70.8 ± 19.9	76.8 ± 22.7	85.8 ± 15.0
	8 wks	90.1 ± 17.6	28.9 ± 11.5 **^α^**	15.7 ± 8.5 **^α^**	18.9 ± 8.7 **^α^**	63.1 ± 10.8	79.4 ± 22.9	81.5 ± 27.2	89.4 ± 24.6
	10 wks	92.6 ± 10.7	36.9 ± 16.4 **^α^**	19.6 ± 12.9 **^α^**	21.9 ± 12.9 **^α^**	65.6 ± 21.1	84.5 ± 16.8	85.4 ± 20.4	82.2 ± 21.8
	12 wks	81.5 ± 25.3	39.8 ± 7.8	16.5 ± 7.8 **^α^**	17.5 ± 10.9 **^α^**	66.3 ± 14.3	78.1 ± 27.9	86.8 ± 23.8	87.7 ± 19.7
PNS index ***^,¥^**	0 wks	4.1 ± 2.3	−3.1 ± 0.7 **^α^**	−4.2 ± 0.5 **^α^**	−3.5 ± 0.8 **^α^**	−1.0 ± 1.2 **^α^**	−0.2 ± 1.6 **^α^**	1.2 ± 1.9	2.1 ± 2.0
	2 wks	4.0 ± 2.0	−3.0 ± 0.8 **^α^**	−4.0 ± 0.6 **^α^**	−3.4 ± 0.6 **^α^**	−0.1 ± 1.4 **^α^**	1.1 ± 2.0 **^α^**	2.5 ± 2.4	2.9 ± 2.3
	4 wks	3.9 ± 2.6	−2.7 ± 0.5 **^α^**	−3.9 ± 0.5 **^α^**	−3.1 ± 0.5 **^α^**	0.4 ± 0.9 **^α^**	2.8 ± 3.6	4.6 ± 4.6	5.0 ± 4.0
	6 wks	4.0 ± 1.8	−2.5 ± 0.7 **^α^**	−3.6 ± 0.6 **^α^**	−3.0 ± 0.5 **^α^**	0.6 ± 0.9 **^α^**	1.2 ± 1.8	1.6 ± 1.3	2.1 ± 0.7
	8 wks	4.0 ± 1.8	−2.8 ± 0.6 **^α^**	−3.6 ± 0.5 **^α^**	−3.0 ± 0.5 **^α^**	0.6 ± 0.9 **^α^**	2.3 ± 1.4 **^α^**	2.2 ± 1.8	3.1 ± 1.6
	10 wks	5.0 ± 2.4	−2.3 ± 0.8 **^α^**	−3.4 ± 0.6 **^α^**	−2.7 ± 1.0 **^α^**	0.8 ± 1.4 **^α^**	2.7 ± 1.9 **^α^**	3.1 ± 1.7	3.3 ± 1.7
	12 wks	4.1 ± 1.2	−2.3 ± 0.7 **^α^**	−3.6 ± 0.6 **^α^**	−2.9 ± 0.9 **^α^**	0.9 ± 1.2 **^α^**	2.2 ± 1.8	2.5 ± 1.5	3.3 ± 1.4

***^,¥^**: significant group-by-time interaction and time effects; **^α^**: within-group differences compared to pre-exercise value; **^x^**: between-group differences at each exercise session compared to weeks 0 and 2, respectively; SD1: standard deviation of the Poincaré plot perpendicular to the line of identity; SD2: standard deviation of the Poincaré plot along the line of identity; PNS: parasympathetic nervous system; wks: weeks.

**Table 7 animals-15-01122-t007:** Changes in the SNS index (mean ± SD) in UAHs and UGHs before, during, and after the exercise bouts at 2-week intervals over the 12-week structured exercise programme.

Groups	Stage	Pre- Exercise	Exercise			Post-Exercise			
			Ex-1	Ex-2	Ex-3	30 min	60 min	90 min	120 min
UAH ***^,#,¥^**	0 wk	−2.5 ± 0.5	4.4 ± 1.9 **^α^**	13.9 ± 1.8 **^α^**	8.4 ± 3.5 **^α^**	−0.5 ± 0.7 **^α^**	−1.3 ± 1.1	−1.6 ± 0.7	−1.7 ± 0.8
	2 wk	−2.1 ± 0.9	3.8 ± 1.2 **^α^**	8.9 ± 2.2 **^α,x^**	6.0 ± 2.9 **^α^**	−1.3 ± 0.7	−2.0 ± 1.0	−2.3 ± 0.6	−2.4 ± 0.5
	4 wk	−1.8 ± 1.5	4.2 ± 1.9 **^α^**	9.4 ± 3.1 **^α,x^**	6.3 ± 3.7 **^α^**	−1.4 ± 0.9	−2.1 ± 0.9	−2.3 ± 0.9	−2.1 ± 0.9
	6 wk	−2.2 ± 0.6	3.2 ± 1.3 **^α^**	8.6 ± 2.7 **^α,x^**	5.1 ± 2.6 **^α^**	−1.0 ± 1.0	−1.7 ± 0.8	−2.2 ± 0.8	−2.2 ± 0.5
	8 wk	−2.5 ± 0.4	3.4 ± 1.2 **^α^**	8.2 ± 2.2 **^α,x^**	5.1 ± 2.0 **^α^**	−1.2 ± 0.9	−1.9 ± 0.7	−2.1 ± 0.7	−2.2 ± 0.7
	10 wk	−2.7 ± 0.5	2.7 ± 1.0 **^α^**	6.5 ± 1.7 **^α,x^**	4.1 ± 2.0 **^α^**	−2.0 ± 1.9	−1.8 ± 0.9	−2.2 ± 0.7	−2.5 ± 0.5
	12 wk	−2.4 ± 0.4	2.7 ± 0.8 **^α^**	7.3 ± 1.7 **^α,x^**	4.2 ± 1.4 **^α^**	−1.3 ± 1.0	−1.7 ± 1.2	−1.9 ± 0.9	−1.9 ± 1.0
UGH **^#,¥^**	0 wk	−2.3 ± 0.7	5.6 ± 2.4 **^α^**	13.9 ± 4.1 **^α^**	10.7 ± 4.6 **^α^**	1.4 ± 2.1 **^α^**	0.2 ± 1.6 **^α^**	−0.9 ± 1.3 **^α^**	−1.5 ± 1.0
	2 wk	−2.4 ± 0.7	5.8 ± 4.2 **^α^**	11.5 ± 4.2 **^α^**	9.1 ± 4.1 **^α^**	−0.03 ± 1.6 **^α^**	−0.9 ± 1.6	−1.7 ± 1.3	−2.1 ± 0.7
	4 wk	−2.3 ± 0.9	4.1 ± 1.5 **^α^**	12.1 ± 7.6 **^α^**	6.9 ± 1.9 **^α^**	−0.5 ± 1.0 **^α^**	−1.7 ± 0.8	−2.2 ± 0.9	−2.5 ± 0.7
	6 wk	−2.5 ± 0.6	3.8 ± 2.4 **^α^**	9.2 ± 3.1 **^α^**	6.5 ± 2.9 **^α^**	−0.8 ± 0.7 **^α^**	−1.1 ± 1.0	−1.4 ± 0.6	−1.8 ± 0.4
	8 wk	−2.4 ± 0.8	4.5 ± 2.9 **^α^**	8.9 ± 2.6 **^α^**	6.6 ± 3.1 **^α^**	−0.9 ± 0.7 **^α^**	−1.9 ± 0.7	−1.8 ± 0.9	−2.2 ± 0.6
	10 wk	−2.8 ± 0.7	3.1 ± 2.1 **^α^**	8.6 ± 4.3 **^α^**	5.9 ± 4.5 **^α^**	−0.8 ± 1.3 **^α^**	−1.9 ± 1.0	−2.1 ± 0.8	−2.2 ± 0.7
	12 wk	−2.6 ± 0.5	2.8 ± 1.6 **^α^**	8.6 ± 3.1 **^α^**	6.4 ± 3.6 **^α^**	−1.1 ± 1.0	−1.7 ± 1.0	−2.0 ± 0.6	−2.3 ± 0.6

***^,#,¥^**: significant group-by-time interaction, group, and time effects; **^α^**: within-group differences compared to pre-exercise value; **^x^**: between-group differences at each exercise session compared to week 0; SNS: sympathetic nervous system.

**Table 8 animals-15-01122-t008:** Changes in the stress index (mean ± SD) in UAHs and UGHs before, during, and after exercise bouts at 2-week intervals over the 12-week structured exercise programme.

Groups	Stage	Pre- Exercise	Exercise			Post-Exercise			
			Ex-1	Ex-2	Ex-3	30 min	60 min	90 min	120 min
UAH ***^,#,¥^**	0 wks	3.9 ± 1.2	14.7 ± 3.2 **^α^**	40.3 ± 9.4 **^α^**	33.5 ± 14.0 **^α^**	7.9 ± 2.5	6.8 ± 2.7	5.6 ± 1.4	5.2 ± 2.0
	2 wks	4.8 ± 1.2	14.7 ± 2.3 **^α^**	28.2 ± 7.4 **^α^**	27.6 ± 9.0 **^α^**	6.8 ± 1.8	5.4 ± 2.5	4.4 ± 1.5	4.1 ± 1.2
	4 wks	5.7 ± 3.6	17.0 ± 7.2 **^α^**	27.0 ± 7.6 **^α^**	26.2 ± 14.0 **^α^**	6.7 ± 2.4	5.6 ± 2.1	5.2 ± 1.8	5.4 ± 1.8
	6 wks	5.6 ± 2.7	13.1 ± 3.5 **^α^**	28.8 ± 7.4 **^α^**	23.6 ± 8.2 **^α^**	7.8 ± 3.1	6.2 ± 1.9	4.8 ± 1.5	5.1 ± 1.1
	8 wks	4.4 ± 0.9	14.3 ± 3.4 **^α^**	26.9 ± 8.9 **^α^**	23.8 ± 5.2 **^α^**	6.9 ± 2.8	5.6 ± 1.9	5.1 ± 1.6	5.0 ± 1.8
	10 wks	3.7 ± 1.1	13.2 ± 1.7 **^α^**	21.3 ± 5.9 **^α,x^**	22.2 ± 8.5 **^α^**	6.9 ± 2.7	6.4 ± 3.3	4.8 ± 1.6	4.5 ± 1.7
	12 wks	4.3 ± 1.2	12.8 ± 3.0 **^α^**	25.3 ± 5.5 **^α,x^**	22.0 ± 5.4 **^α^**	7.8 ± 3.2	6.3 ± 3.1	5.4 ± 2.0	5.6 ± 2.3
UGH ***^,#,¥^**	0 wks	4.2 ± 1.7	18.9 ± 7.7 **^α^**	44.1 ± 13.5 **^α^**	42.2 ± 11.7 **^α^**	13.2 ± 6.5 **^α^**	9.9 ± 4.6 **^α^**	7.4 ± 2.6 **^α^**	5.7 ± 1.7
	2 wks	4.3 ± 1.4	19.8 ± 9.5 **^α^**	36.2 ± 11.6 **^α^**	35.6 ± 13.7 **^α^**	8.9 ± 3.5 **^α^**	7.6 ± 2.9	5.3 ± 2.2	4.5 ± 1.5
	4 wks	5.0 ± 2.0	16.7 ± 3.8 **^α^**	32.3 ± 11.0 **^α^**	28.7 ± 7.3 **^α^**	8.2 ± 3.0 **^α^**	5.3 ± 1.8	4.8 ± 2.1	4.1 ± 1.7
	6 wks	4.5 ± 1.6	16.5 ± 6.1 **^α^**	33.1 ± 10.5 **^α^**	27.2 ± 12.3 **^α^**	7.1 ± 1.3 **^α^**	6.8 ± 1.2	6.0 ± 1.3	4.9 ± 1.0
	8 wks	4.4 ± 1.9	17.7 ± 8.4 **^α^**	30.8 ± 8.7 **^α^**	27.8 ± 13.1 **^α^**	7.2 ± 0.8 **^α^**	5.3 ± 1.2	5.2 ± 1.2	4.9 ± 1.1
	10 wks	3.6 ± 1.1	13.1 ± 3.7 **^α^**	31.4 ± 13.8 **^α^**	24.6 ± 12.1 **^α^**	7.1 ± 2.3 **^α^**	5.0 ± 1.2	4.9 ± 1.7	5.0 ± 1.6
	12 wks	4.5 ± 0.6	11.9 ± 2.8 **^α^**	29.1 ± 7.7 **^α^**	26.5 ± 10.2 **^α^**	6.2 ± 1.8	5.5 ± 1.7	4.9 ± 0.9	4.5 ± 0.9

***^,#,¥^**: significant group-by-time interaction, group, and time effects; **^α^**: within-group differences compared to pre-exercise value; **^x^**: between-group differences at each exercise session compared to week 0; wks: weeks.

## Data Availability

The original data presented in this article are openly available at https://www.doi.org/10.6084/m9.figshare.28330934.

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
