# Peer review of "Dynamic Adaptation of Heart Rate and Autonomic Regulation During Training and Recovery Periods in Response to a 12-Week Structured Exercise Programme in Untrained Adult and Geriatric Horses"

_animals, 2025, doi:10.3390/ani15081122_

Round 1
Reviewer 1 Report
Comments and Suggestions for Authors
This is an interesting manuscript and worthy of publication in this journal.
Line 36 UAHs here should be UGHs.
Line 39, 41 "faster" statistically? Were any statistics performed? No statistics are described in the abstract.
Line 94, 97 Is it possible that the ages are switched between the two groups here? It says UAH horses were 20, UGH were 8. Isn't that backwards? Is the other identifying information correct or switched inadvertently?
Line 169 Authors should mention statistical methods in the abstract.
Line 195 Are there other studies in horses which examined recovery HR responses to training? This current submission is a significant data set but this recovery portion of it seems not to have been discussed in much detail, but typically is more evident of training responses than exercise HR itself.
Line 341 But many more studies show no change with resting HR and training in horses, and even those that show a change show a fractional change which is clinically insignificant.
Line 355 Do the authors have an opinion about how much of this exercising HR change is physiological (training adaptation) and how much is psychological (relaxing, mental adjustment to the new workload)?
Author Response
This is an interesting manuscript and worthy of publication in this journal.
Dear Reviewer 1
I’d like to thank the reviewer’s time and dedication in reviewing my work. I’ve addressed all points and the revisions are highlighted in green within the main text.
Line 36 UAHs here should be UGHs.
Response to the reviewer
I’ve revised it on page 1, line 36, accordingly.
Line 39, 41 "faster" statistically? Were any statistics performed? No statistics are described in the abstract.
Response to the reviewer
Due to the limited word count (200 words) in the abstract, I could only give a summarised result of each experimental section. Regarding the reviewer’s concern, it means that the duration of decreased HRV during exercise sessions reduced (shortened in other words) gradually in response to the 12-week exercise programme in both adult and aged horses. However, based on the results in Tables 2, 7 and 8, the decrease in duration of elevated HRV metrics was faster in adult horses than in aged horses, including HR (2-week vs. 4-week), SNS index (2-week vs 12-week) and stress index (limited to ex-3 vs 12-week).
Line 94, 97 Is it possible that the ages are switched between the two groups here? It says UAH horses were 20, UGH were 8. Isn't that backwards? Is the other identifying information correct or switched inadvertently?
Response to the reviewer
I do apologise for this serious mistake. I’ve switched the age of horses to the group where they belong on page 3, lines 97 and 100, accordingly.
Line 169 Authors should mention statistical methods in the abstract.
Response to the reviewer
I’ve added the statistical analysis method in the abstract on page 1, lines 37-38.
Line 195 Are there other studies in horses which examined recovery HR responses to training? This current submission is a significant data set but this recovery portion of it seems not to have been discussed in much detail, but typically is more evident of training responses than exercise HR itself.
Response to the reviewer
Based on the reviewers' queries, I found that several studies have reported on post-exercise heart rate recovery (https://doi.org/10.1111/j.2042-3306.2010.00260.x, https://doi.org/10.3390/ani10010120, https://doi.org/10.1111/evj.12267_19). However, examining the heart rate recovery response is not the main objective of this study. Our previous research on heart rate (HR) and heart rate variability (HRV) at rest was unable to quantify the effectiveness of the exercise program, as there were no changes in HRV metrics due to vagal saturation at rest (https://doi.org/10.1016/0165-1838(96)00028-8, https://doi.org/10.1016/j.jevs.2024.105094). Therefore, I conducted the current study to determine whether HR and HRV adaptations can be quantified by measuring these parameters during exercise sessions and recovery periods. The results of this study confirm that HR and HRV adaptations can indeed be observed when measured during both the exercise session and the recovery period.
Line 341 But many more studies show no change with resting HR and training in horses, and even those that show a change show a fractional change which is clinically insignificant.
Response to the reviewer
I agree with the reviewer that some studies may report no change in resting HR and training in horses. In this study, the resting HR measured before the exercise seems to be reduced at 2-week intervals across the 12-week exercise programme based on the mean value of HR in adult horses (41.2 at the first exercise program to 39.9 bpm at the 12-week exercise programme) and aged horses (42.8 at the first exercise program to 37.0 bpm at the 12-week exercise programme). However, the statistical tests showed no significance across the 12-week exercise period in both groups. That’s why we suggested on page 13, lines 344-352, that the lack of a significant change in resting HR may be attributed to “individual variations” as it is reported to be substantial in horses (https://doi.org/10.1007/s00114-022-01811-y, https://doi.org/10.1111/evj.13017). This variation could underlie the observed inconsistencies within each group, mainly since they are untrained horses.
Line 355 Do the authors have an opinion about how much of this exercising HR change is physiological (training adaptation) and how much is psychological (relaxing, mental adjustment to the new workload)?
Response to the reviewer
I believe that the change in heart rate (HR) in response to the proposed exercise program is primarily due to physiological adaptation (training adaptation), as horses were given the same management, including but not limited to diet and housing, as stated on page 3, lines 104-107. Furthermore, they are undergoing the same exercise programme, which includes a consistent activity pattern and duration. A decrease in HR with the same exercise indicates that training is likely increasing stroke volume. The increased stroke volume, a lower HR, allows the heart to maintain adequate cardiac output for the body's needs. However, we should not overlook the potential impact of psychological adaptation, and further research is needed for a deeper investigation.

Reviewer 2 Report
Comments and Suggestions for Authors
animals 3491324
Dynamic Adaptation of Heart Rate and Autonomic Regulation During Training and Recovery Periods in Response to a 12-week Structured Exercise Programme in Untrained Adult and Geriatric Horses
This paper presents the results of a study in which heart rate and measures of heart rate variability were examined during a 12 week exercise program in 2 separate groups of horses. While the objective appears to have been to determine how heart rate and heart rate variability changed, the authors also comment on the effect of age, and simultaneously on the effect of previous training, and hypothesize that response patterns would differ between horses with different exercise histories. The objectives are in fact somewhat unclear and are not clearly stated, while the experimental design and choice of subjects does not appear to be optimal. In the process of proceeding without clearly defined objectives and appropriate experimental design, the authors have impacted their ability to ask questions of their data because of confounding.
This lack of focus extends to the choice of HRV indices to use to assess effects and creates an unnecessary degree of study complexity. The investigation would benefit from a narrower and tighter focus, from a more targeted choice of outcomes, and an attempt to answer fewer and more specific questions. Comparison between the two groups is not really possible because of the confounding, and with appropriate constraints, the study might be better divided into two papers with each paying appropriate attention to limitations. It is apparent that the authors have put a lot of effort into this investigation, and their efforts need to be reflected in the quality of presentation, that is if the methodological and structural issues can be resolved. My concerns are as follows, not necessarily in any particular order.
I have difficulty with the idea of the groups being "untrained adult" and "geriatric", since there appear to be more older horses in the adult group than in the geriatric group. The age range for the adult horses suggests that these are in fact the geriatric horses, since if the authors' assertions are correct, they have reached an advanced age without ever having done any work. I also don't understand how the mean age of these horses could be 8.8+ or -2 years yet they have been retired from competition at around 15 years of age?. Insufficient information is presented on their histories. Bearing in mind they represent members of a laboratory herd, it is possible that they may have been used in the past for a range of non-equitation purposes. The group described as geriatric appear to be significantly younger, to have had a role in equitation possibly for some years, and some may continue to be in light work.The authors also provide insufficient information on selection criteria. It might have been better to simply generate a binary variable to describe each horse's exercise status, record their age, and whether each was currently in any sort of work, and study all horses under identical conditions as one group. As it is, the information provided and the study design used make this impossible and the results of the study unsafe.
The two citations referenced as representing validations of the equipment employed, and by inference the signal collected and its technical parameters, are not in fact full validations. Validation requires a focused investigation in which input and output generated by a test system are compared directly with the same input and output captured using a validated or gold standard method and the two sets of output compared. The simple fact the equipment has previously been used in published studies does not necessarily constitute validation. Also, the studies all used different methodologies and measures - their results cannot be applied to HRV analysis in general and only examine the time domain whereas the authors of this study attempt to report on all domains.
Line 143. These procedures are problematic. They can only be used in support of data quality and consistency if it is known precisely what adjustments software made to each recording and that that process was the same for all horses. For example, were the raw recordings manually examined for quality and consistency and errors classified and categorized, or were the recordings simply gathered and subjected to whatever processing the software decided needed to be employed? To state this in other words, was the processing of the files essentially automated using a "black box" approach? Reference is made to corrected and uncorrected data, yet the specifics are not provided to allow the reader to assess the validity of the approach.
Line 143. Please provide more information on the quality of the recordings. If specific sections were taken at half hour intervals, what was the quality, how often were data missing, and what is the number for each of the collection intervals - this information is not provided in any of the tables.
Line 149. Please provide additional information on this procedure, particularly the software that was employed and potential impacts on the data and results. Was this technique employed consistently on all tracing or only on those for which you detected "non-stationarities"? What would have been the impact of applying both the smoothness priors technique and the 0.035 Hz cutoff to the same data? Do the authors have information on precisely what mathematical procedures were employed and how these influenced data content?
Line 150. Please explain how the technique of employing a 0.035 Hz cut-off influenced the data and your frequency bands. Was this employed in the Polar recording device or was this applied in the Kubios software post-collection? Was the technique applied only to the entire recording or was it applied separately to sections of tracing that were used, post-collection?
Table 2. What portion of the effect observed over time might have been due to familiarisation with the procedure?
Line 165. Were data tested for normality?
Table 2. The standard deviation during exercise 2 and 3 reveals a pattern suggesting that there might've been significant variation in horses' level of fitness prior to the onset of the training regimen. How was this examined?
Table 2. For group by time interaction, are you referring to stage or successive sampling windows (half hour intervals) as time, or both? One of the greatest difficulties I have with this study is the ambiguity involved when the authors refer to "time". There are in fact 2 axes of time involved, time over the training program of 12 weeks and time within a training session of 120 minutes. Trying to differentiate between these two when reading the text or viewing the tables is horrendously time-consuming and rarely clear. If the authors were to do nothing else with this paper, this issue needs to be resolved. It becomes particularly difficult when attempting to interpret the results. The authors could refer, for example, to T-time indicating training response over the 12 week study, and E-time indicating elapsed time within an exercise from 0-120. Alternatively reference could be made to study time to indicate the totality of the study and exercise time to reference elapsed time within an exercise session. Whichever, something must be done about this to make the authors' hard work accessible. At present, it isn't even clear what is meant by group-by-time interaction.
Line 292. Please define what you mean by SNS index and PNS index and confirm that these measures have been independently validated. The ready availability of these numbers in published software does not necessarily mean that they are scientifically valid. In particular, note that these parameters are calculated in the software by making assumptions about "normal" population levels, something that has not yet been reliably determined for horses!
The paper seems to take a "weight of evidence" approach by employing every published indicator of autonomic nervous system activity and others that are not yet examined in the peer—reviewed liiterature, yet so many of these indices are interdependent. In this context, variables referenced in the KUBIOS software but not independently verified are used as though validated. The paper is exceedingly complex and the authors are not effective in presenting or summarizing the information so as to help the reader follow what is going on. Presentation of data could be helped considerably by using graphs to present findings. Raw results could (must also) be presented in supplementary tables. The average reader could then understand the broad messages, referring to the supplemental information for support.
I wonder whether the exercise program employed here was actually sufficient to elicit a reliable response. This concern arises in part because of the lack of adequate information on the exercise the horses were undertaking during their routine maintenance program prior to the study as well as during the investigation. If we assume that no interference could have arisen in this regard, we also need to be confident that the period immediately preceding the study was sufficiently uniform and long that any prior training effect would have subsided. Reassurance that the study population was homogeneous in this regard is not provided.
Although the structured exercise is described in terms of pattern, no information is provided on speed and how consistently this was maintained. The facility used for the exercise is described as a round pen, but no information is provided otherwise. How large was the pen, what means did the person supervising the exercise employ to encourage horses to maintain pace, what was the footing. Where were electrodes positioned on the horse?
The decision to standardize and hold constant the pattern by which individual horses were presented for work and data collection is (line 134) highly problematic because it introduces an additional variable, order of exercise. Order should have been randomized. It is quite possible that this order and the resultant time of day in relation to routine stable procedures such as feeding had an effect on the outcomes. The researchers must introduce a variable that reflects this order and include that in their analyses to deal with this problem statistically, testing the data very carefully to ensure that there is no confounding with other variables. For example, what was the relationship between order of presentation for exercise and age - if it transpired that the oldest horses were always done earlier in the day than the younger horses, these two variables cannot be separated in analysis since the source of any effect cannot be resolved.
The authors appear to have discounted the significance of the "geriatric" group's previous exercise history. Because of a host of impacts, including work-associated injuries, previous response to training and their capacity, therefore, to respond to the light training exercise, there becomes an additional possible explanation for the observations made.
Notwithstanding the authors reference to a literature that supports the use of the equipment and procedures employed in this study, it should be noted that those validations are in reference to specific measures, and each study used different procedures. This validation process is thus currently incomplete and does not cover the range of heart rate variability parameters employed by the current authors. Despite this, the authors present a very broad suite of parameters and assume their validity in the horse. The authors have included primarily supportive references in their bibliography and might be somewhat selective in their choice of findings to quote. A more circumspect interpretation of findings might be advisable, while the paper could be considerably simplified and strengthened by using only those parameters that have been reliably validated.
Author Response
Dynamic Adaptation of Heart Rate and Autonomic Regulation During Training and Recovery Periods in Response to a 12-week Structured Exercise Programme in Untrained Adult and Geriatric Horses
Dear Reviewer 2
First of all, I’d like to thank the reviewer’s time and dedication in reviewing my work. I’ve addressed all points and the revisions are highlighted in green within the main text. The context that already mentioned was highlighted in yellow.
This paper presents the results of a study in which heart rate and measures of heart rate variability were examined during a 12 week exercise program in 2 separate groups of horses. While the objective appears to have been to determine how heart rate and heart rate variability changed, the authors also comment on the effect of age, and simultaneously on the effect of previous training, and hypothesize that response patterns would differ between horses with different exercise histories. The objectives are in fact somewhat unclear and are not clearly stated, while the experimental design and choice of subjects does not appear to be optimal. In the process of proceeding without clearly defined objectives and appropriate experimental design, the authors have impacted their ability to ask questions of their data because of confounding.
Response to the reviewer
I would like to thank the reviewer for the feedback and agree that the objective and hypothesis statement was unclear. I have revised the rationale, objective, and hypothesis to provide more transparent information that aligns with the experimental design on page 2, lines 82-91.
This lack of focus extends to the choice of HRV indices to use to assess effects and creates an unnecessary degree of study complexity. The investigation would benefit from a narrower and tighter focus, from a more targeted choice of outcomes, and an attempt to answer fewer and more specific questions. Comparison between the two groups is not really possible because of the confounding, and with appropriate constraints, the study might be better divided into two papers with each paying appropriate attention to limitations. It is apparent that the authors have put a lot of effort into this investigation, and their efforts need to be reflected in the quality of presentation, that is if the methodological and structural issues can be resolved. My concerns are as follows, not necessarily in any particular order.
Response to the reviewer
I’d like to thank the reviewer’s feedback. Please allow me to explain what I have done for this study. I’ve conducted an experiment to test whether or not the structured exercise programme can improve HR and autonomic regulation by showing the modulating changes in those parameters. Despite improving autonomic regulation, defined by increased resting HRV (https://doi.org/10.1038/s41598-025-86679-4), there was no modulating change in the autonomic regulation when measuring “resting HRV metrics” in response to the 12-week structured exercise programme due to full vagal saturation at rest in horses (https://doi.org/10.1016/0165-1838(96)00028-8, https://doi.org/10.1371/journal.pone.0259933). For this reason, I later conducted the current experiment to test whether the modulating changes in HRV metrics can be found when measuring “during the exercise session and recovery period” rather than at rest. Moreover, as the modulating changes were expected to be observed, the effectiveness of using this measurement in horses receiving similarly structured exercise programmes may be estimated if we conducted the experiment in distinct horse groups with the same fitness level (untrain condition) and management condition but different ages (adult vs ageing). In fact, there were no genuine statistical comparisons between the two groups of horses in this study. Instead, I compared the “trend of changes” in HR and HRV metrics when two groups received the same exercise regimen and measured at a similar period during exercise sessions and recovery. In the present study, HR and HRV metrics were adapted in response to both groups' 12-week structure exercise programme. Moreover, modulating changes in HR and HRV metrics in the untrained adult horse were noticed earlier (reflecting faster adaptation) than those in untrained aged horses (2-4 weeks in untrained adult horses vs. 10-12 weeks in untrained aged horses) for example.
Regarding the choice of HRV indices, I primarily selected the variable of each analysis method as it can indicate sole vagal activity or combined vagal and sympathetic activities, including time-domain method: RR interval, SDNN and RMSSD; frequency domain method: LF, HF and LF/HF ratio. Although the different extent of changes in these HRV metrics was found between the two groups, the extent of adaptations cannot be quantified by these metrics. That’s why we additionally selected the other HRV metrics, including time-domain variable SDANN, SDNNI, and ANS index: PNS and SNS index, to quantify the extent of adaptation in this study. The benefit of adopting SDANN and SDNNI in this study has been discussed on page 14, lines 383-388. Nonlinear variable SD1, SD2 and SD2/SD1 ratio were additionally measured due to the fact that the complexity of cardiac pattern may be observed as the extent of adaptation cannot measured by time and frequency domain analysis (https://doi.org/10.1016/j.physbeh.2007.01.007).
I hope that this explanation will provide more transparent information and convince the reviewer of this study's advantages in the equine field.
I have difficulty with the idea of the groups being "untrained adult" and "geriatric", since there appear to be more older horses in the adult group than in the geriatric group. The age range for the adult horses suggests that these are in fact the geriatric horses, since if the authors' assertions are correct, they have reached an advanced age without ever having done any work. I also don't understand how the mean age of these horses could be 8.8+ or -2 years yet they have been retired from competition at around 15 years of age?. Insufficient information is presented on their histories. Bearing in mind they represent members of a laboratory herd, it is possible that they may have been used in the past for a range of non-equitation purposes. The group described as geriatric appear to be significantly younger, to have had a role in equitation possibly for some years, and some may continue to be in light work.The authors also provide insufficient information on selection criteria. It might have been better to simply generate a binary variable to describe each horse's exercise status, record their age, and whether each was currently in any sort of work, and study all horses under identical conditions as one group. As it is, the information provided and the study design used make this impossible and the results of the study unsafe.
Response to the reviewer
I sincerely apologize for the serious error regarding the age range and study groups mentioned in the Materials and Methods section. Specifically, the study included nine untrained adult horses (UAHs)—four geldings, four mares, and one stallion—aged 8.8 ± 2.0 years and weighing 306.4 ± 75.8 kg. Additionally, it comprised nine untrained geriatric horses (UGHs)—five geldings and four mares—aged 20.8 ± 3.9 years and weighing 371.1 ± 51.1 kg. I have made the necessary revisions on page 3, lines 97 and 100.
The two citations referenced as representing validations of the equipment employed, and by inference the signal collected and its technical parameters, are not in fact full validations. Validation requires a focused investigation in which input and output generated by a test system are compared directly with the same input and output captured using a validated or gold standard method and the two sets of output compared. The simple fact the equipment has previously been used in published studies does not necessarily constitute validation. Also, the studies all used different methodologies and measures - their results cannot be applied to HRV analysis in general and only examine the time domain whereas the authors of this study attempt to report on all domains.
Response to the reviewer
I would like to express my gratitude for the reviewer’s feedback regarding the validation issue. The use of a heart rate monitor (HRM) device for measuring time and frequency domain variables has been validated in horses (see https://doi.org/10.1016/j.applanim.2021.105401). I have cited this article in the methods section to enhance the transparency of the study. The reviewer might argue that other variables, such as nonlinear measures and the autonomic nervous system (ANS) index, have not yet been validated in horses. However, the current study does not aim to provide reference heart rate variability (HRV) values for comparison in horses in general. Instead, I have adopted potential variables from the validated HRV analysis program (see https://www.kubios.com/hrv-scientific/) and compared them within this study to assess the effect of the exercise programme on horses of different ages. I ensured that the analysis conditions remained consistent, including device settings, the HRV analysis program, and correction for beats and noise among horses undergoing a similar exercise program over a similar period. I believe that changes in HRV variables could indicate the degree of HRV modulation and adaptation, thus providing sufficient data for comparison and being beneficial to the equine field.
Line 143. These procedures are problematic. They can only be used in support of data quality and consistency if it is known precisely what adjustments software made to each recording and that that process was the same for all horses. For example, were the raw recordings manually examined for quality and consistency and errors classified and categorized, or were the recordings simply gathered and subjected to whatever processing the software decided needed to be employed? To state this in other words, was the processing of the files essentially automated using a "black box" approach? Reference is made to corrected and uncorrected data, yet the specifics are not provided to allow the reader to assess the validity of the approach.
Response to the reviewer
I would like to express my gratitude for the reviewer’s comments. I have adopted the Kubios HRV Scientific software (https://www.kubios.com/hrv-scientific/), which offers several additional features compared to the standard version. One complex issue in HRV analysis is artefact correction, as it can significantly distort the results obtained from the HRM device. The correction level should be tailored to each individual, as there are substantial inter-individual differences in HRV. Therefore, a fixed threshold does not work optimally for all subjects (https://www.kubios.com/downloads/Kubios_HRV_Users_Guide.pdf). I am concerned that manual artefact correction may introduce inaccuracies due to human error. For this reason, I chose to use the Kubios HRV Scientific software, which provides automatic artefact correction—an approach that is more accurate and validates the method (https://doi.org/10.1080/03091902.2019.1640306).
According to the guideline, the threshold-based artefact correction algorithm compares each inter-beat interval (IBI) value against a local average interval. This local average is obtained through median filtering of the IBI time series, ensuring that it is not influenced by single outliers. If an IBI deviates from the local average by more than a specified threshold, it is identified as an artefact and marked for correction.
In contrast, the automatic artefact correction algorithm detects artefacts from the dRR series, which consists of the differences between successive RR intervals. The dRR series offers a robust method for distinguishing ectopic and misplaced beats from the normal sinus rhythm, which can vary between subjects. A time-varying threshold (Th) is used to differentiate ectopic from normal beats. This threshold is estimated from the time-varying distribution of the dRR series to adapt to different HRV levels. For each beat, the quartile deviation of the 90 surrounding beats is calculated and multiplied by a factor of 5.2. This method captures 99.95% of all beats when the RR series is normally distributed. However, since the RR interval series is often not normally distributed, some normal beats may still exceed the threshold. Thus, a decision algorithm is essential for detecting artefact beats. Furthermore, this algorithm also provides automatic noise detection, which is particularly useful for analysing long-term RR recordings, as signal quality can be variable. Segments identified as noise are automatically excluded from HRV analysis.
For these reasons, I am confident that using the automatic algorithms will yield accurate data and produce reliable results in this study.
Line 143. Please provide more information on the quality of the recordings. If specific sections were taken at half hour intervals, what was the quality, how often were data missing, and what is the number for each of the collection intervals - this information is not provided in any of the tables.
Response to the reviewer
As described on page 4, lines 138-140 and 167-169, RR interval data were continuously collected starting 30 minutes before the exercise session, throughout the 54-minute exercise bout, and continuing into a 120-minute recovery period. During the exercise session, some heart rate (HR) and heart rate variability (HRV) data were missed due to the program's algorithm flagging excessive artefact and noise. I ensured that the autonomic artefact and noise correction provided by the Kubios program would adequately address ectopic or misaligned beats, as well as any noise that could distort the HRV analysis results. With only one missing data point, GraphPad Prism (version 10.4.1), the software I used for statistical analysis, did not allow for a two-way ANOVA and recommended using a mixed-effects model, as outlined in the statistical analysis section.
Line 149. Please provide additional information on this procedure, particularly the software that was employed and potential impacts on the data and results. Was this technique employed consistently on all tracing or only on those for which you detected "non-stationarities"? What would have been the impact of applying both the smoothness priors technique and the 0.035 Hz cutoff to the same data? Do the authors have information on precisely what mathematical procedures were employed and how these influenced data content?
Response to the reviewer
I would like to provide more clarification on this issue. Kubios HRV software incorporates an advanced detrending method based on smoothness priors regularisation, making it suitable for more complex trends in time series data. This method functions like a time-varying high-pass filter, smoothing the data based on the value of the smoothing parameter. “As the smoothing parameter increases, the cutoff frequency of the filter decreases”. The user guidelines recommend using the smoothness priors method to eliminate nonstationarities in the IBI time series, which are detected in all traces. Only stationary segments are analysed. To ensure that normal short-term heart rate variability is preserved, the cut-off frequency should remain below the low-frequency band threshold of 0.04 Hz (https://www.kubios.com/downloads/Kubios_HRV_Users_Guide.pdf ). That’s why I used the smoothness priors and set the cut-off frequency as lowest as 0.035 Hz in this study.
Line 150. Please explain how the technique of employing a 0.035 Hz cut-off influenced the data and your frequency bands. Was this employed in the Polar recording device or was this applied in the Kubios software post-collection? Was the technique applied only to the entire recording or was it applied separately to sections of tracing that were used, post-collection?
Response to the reviewer
As described above, The user guidelines recommend using the smoothness priors method to eliminate nonstationarities in the IBI time series, which are detected in all traces. Only stationary segments are analysed. To ensure that normal short-term heart rate variability is preserved, the cutoff frequency should remain below the low-frequency band threshold of 0.04 Hz (https://www.kubios.com/downloads/Kubios_HRV_Users_Guide.pdf ). So, the use of 0.035 Hz as a cut-off frequency is according to the user guideline.
Table 2. What portion of the effect observed over time might have been due to familiarisation with the procedure?
Response to the reviewer
Although the familiarisation with the procedure may not be overlooked in the change of HR in this study, in my opinion, it plays a little role in this HR modulation. Due to the evidence of a significant decrease in duration of elevated HR as well as a significant reduction in HR during ex-1, 2 and 3 in UAH and ex-1 and 2 overtime during a 12-week structured exercise, it could be summarised that the changes in HR primarily occur as a result of increased fitness and improve autonomic regulation following the exercise programme.
Line 165. Were data tested for normality?
Response to the reviewer
I have not tested the normality of the data in this study. However, The Greenhouse-Geisser correction was implemented to address the lack of sphericity in the repeated measures ANOVA (or mixed-effects model). This correction serves as both an estimate of epsilon (sphericity) and a mechanism to adjust for the lack of sphericity, ensuring equal variability of differences, as stated on page 4, lines 173-174.
(https://www.graphpad.com/guides/prism/latest/statistics/stat_sphericity_and_compound_symmet.htm).
Table 2. The standard deviation during exercise 2 and 3 reveals a pattern suggesting that there might've been significant variation in horses' level of fitness prior to the onset of the training regimen. How was this examined?
Response to the reviewer
I would like to express my gratitude to the reviewer for their insightful comments. Research indicates that horses exhibit a significant degree of individual variation (https://doi.org/10.1007/s00114-022-01811-y, https://doi.org/10.1111/evj.13017). Consequently, this individual variability is unavoidable, even among horses of similar age groups. I concur with the reviewer that the higher standard deviation (SD) observed in both the UAH and UGH groups at the onset of exercise is likely due to differences in fitness levels or individual variability. In this study, I did not assess the fitness levels of each horse; instead, I used the heart rate variability (HRV) values at week 0 as a control to estimate the adaptation of those variables. The SD appeared to decrease over time in both groups as they engaged in the exercise programme. For instance, in the UAH group, the SD of heart rate (HR) during Ex-3 decreased from 14.1 at week 0 to 9.9 at week 12. Similarly, in the UGH group, the SD decreased from 29.1 to 20.1 by week 12. The potential for individual adaptation may also be a contributing factor to the variation in HR observed in this study. Nonetheless, this trend suggests that the personal fitness levels in both groups were gradually aligning as the training progressed.
Table 2. For group by time interaction, are you referring to stage or successive sampling windows (half hour intervals) as time, or both? One of the greatest difficulties I have with this study is the ambiguity involved when the authors refer to "time". There are in fact 2 axes of time involved, time over the training program of 12 weeks and time within a training session of 120 minutes. Trying to differentiate between these two when reading the text or viewing the tables is horrendously time-consuming and rarely clear. If the authors were to do nothing else with this paper, this issue needs to be resolved. It becomes particularly difficult when attempting to interpret the results. The authors could refer, for example, to T-time indicating training response over the 12 week study, and E-time indicating elapsed time within an exercise from 0-120. Alternatively reference could be made to study time to indicate the totality of the study and exercise time to reference elapsed time within an exercise session. Whichever, something must be done about this to make the authors' hard work accessible. At present, it isn't even clear what is meant by group-by-time interaction.
Response to the reviewer
I do apologise for giving unclear information. The group meant the elapsed time points within a training session of 120 minutes. Whereas, time referred to periods of 12-week training programme. I’ve added this description in the statistical analysis section on page 4, lines 174-176.
Line 292. Please define what you mean by SNS index and PNS index and confirm that these measures have been independently validated. The ready availability of these numbers in published software does not necessarily mean that they are scientifically valid. In particular, note that these parameters are calculated in the software by making assumptions about "normal" population levels, something that has not yet been reliably determined for horses!
Response to the reviewer
I’d like to thank the reviewer for this valuable comment. As stated previously, the reviewer might argue that other variables, such as the autonomic nervous system (ANS) index, have not yet been validated in horses. However, these variables were calculated based on the validated variables. For example, the PNS index is calculated from RR intervals, RMSSD and SD1. On the other hand, mean HR, stress index and SD2 are used to compute the SNS index (https://www.kubios.com/blog/hrv-ans-function/). Moreover, the current study does not aim to provide reference heart rate variability (HRV) values for comparison in horses in general. Instead, I have adopted potential variables from the validated HRV analysis program (see https://www.kubios.com/hrv-scientific/) and compared them within this study to assess the effect of the exercise program on horses of different ages. I ensured that the analysis conditions remained consistent, including device settings, the HRV analysis program, and correction for beats and noise among horses undergoing a similar exercise program over a similar period. I believe that changes in HRV variables could indicate the degree of HRV modulation and adaptation, thus providing sufficient data for comparison and being beneficial to the equine field.
The paper seems to take a "weight of evidence" approach by employing every published indicator of autonomic nervous system activity and others that are not yet examined in the peer—reviewed liiterature, yet so many of these indices are interdependent. In this context, variables referenced in the KUBIOS software but not independently verified are used as though validated. The paper is exceedingly complex and the authors are not effective in presenting or summarizing the information so as to help the reader follow what is going on. Presentation of data could be helped considerably by using graphs to present findings. Raw results could (must also) be presented in supplementary tables. The average reader could then understand the broad messages, referring to the supplemental information for support.
Response to the reviewer
As mentioned earlier, the time-domain and frequency-domain variables recorded from the HRM devices have been validated in horses (https://doi.org/10.1016/j.applanim.2021.105401). Although the PNS and SNS indices have not yet been validated, they are calculated based on the validated variables provided by the program. Furthermore, these variables were analysed and compared within the study, where the horses experienced similar exercise conditions throughout. Despite the need for further research to validate their use and establish reference values for comparison with other studies, these variables still serve to measure modulation and adaptation in this context. In response to the reviewer’s query regarding the presentation of data in graphical format, I have attempted to use graphs to illustrate the study results.
However, while the graph shows the trend of changes in the variables, it isn't easy to indicate statistically significant differences. Therefore, I decided to present the study results more effectively using tables.
I wonder whether the exercise program employed here was actually sufficient to elicit a reliable response. This concern arises in part because of the lack of adequate information on the exercise the horses were undertaking during their routine maintenance program prior to the study as well as during the investigation. If we assume that no interference could have arisen in this regard, we also need to be confident that the period immediately preceding the study was sufficiently uniform and long that any prior training effect would have subsided. Reassurance that the study population was homogeneous in this regard is not provided.
Response to the reviewer
I would like to thank the reviewer for their comments. Both adult and geriatric horses were truly untrained and spent their sedentary lifestyle, as stated on page 3, lines 101-105. This notion confirms neither the effect of the previous training programme nor the current exercise activity was observed before the study commenced. This study found a progressive shortening of the duration of HRV modulation in response to the 12-week training programme. Additionally, I observed a significant decrease in heart rate during Ex-1, 2, and 3 in the UAH group, as well as during Ex-1 and 2 in the UGH group (see Table 2). Furthermore, there was a reduction in the stress index and sympathetic nervous system (SNS) index during Ex-2 in UAH but not in UGH (refer to Tables 7 and 8). These results support the idea that the exercise program provided sufficient stimuli to facilitate HR and HRV adaptation in horses throughout the study.
Although the structured exercise is described in terms of pattern, no information is provided on speed and how consistently this was maintained. The facility used for the exercise is described as a round pen, but no information is provided otherwise. How large was the pen, what means did the person supervising the exercise employ to encourage horses to maintain pace, what was the footing. Where were electrodes positioned on the horse?
Response to the reviewer
Unfortunately, we were unable to measure the speed and distance of the exercise because we could not install the measuring device on the horses during the exercise sessions, which involved round-pen lunging without the use of lunging aids or riders. As a result, our study does not provide information on the average speed and distance covered by the horses during these sessions. This issue has been mentioned in the limitations section on page 15, lines 419-423. Despite the lack of speed data, the horses followed consistent exercise patterns (walking, trotting, and cantering) for durations that allowed them to move within a limited speed range for each gait. So, it was assumed that each horse ran at almost the same speed and duration throughout the 12-week exercise period. Additionally, the exercise was supervised by two skilled lungers who were responsible for maintaining control over the horses' speed throughout each exercise pattern. I have included details about the round pen size, the lunging surface, and the lungers on page 3, lines 120-125. The electrode and sensor pocket were placed on the left side of the horses' chests, as noted on page 4, lines 136-137.
The decision to standardize and hold constant the pattern by which individual horses were presented for work and data collection is (line 134) highly problematic because it introduces an additional variable, order of exercise. Order should have been randomized. It is quite possible that this order and the resultant time of day in relation to routine stable procedures such as feeding had an effect on the outcomes. The researchers must introduce a variable that reflects this order and include that in their analyses to deal with this problem statistically, testing the data very carefully to ensure that there is no confounding with other variables. For example, what was the relationship between order of presentation for exercise and age - if it transpired that the oldest horses were always done earlier in the day than the younger horses, these two variables cannot be separated in analysis since the source of any effect cannot be resolved.
Response to the reviewer
I would like to thank the reviewer for this comment. Two lungers were recruited to exercise the horses in both groups. One lunger was responsible for the horses in the UAH group, while another lunger handled the horses in the UGH group throughout the 12-week study period. To ensure consistency across exercise sessions, the same group of horses was recorded in the same order each time. As previously reported, heart rate variability (HRV) variables can change due to circadian rhythms (https://doi.org/10.1111/j.2042-3306.1999.tb05213.x, https://doi.org/10.1016/j.applanim.2015.02.007). Therefore, the time of day is a crucial factor that should be controlled during HRV analysis. Thus, this approach aims to maintain a similar effect of circadian rhythms—though it may play a minor role—on the HRV modulation of each horse throughout the 12-week study period.
The authors appear to have discounted the significance of the "geriatric" group's previous exercise history. Because of a host of impacts, including work-associated injuries, previous response to training and their capacity, therefore, to respond to the light training exercise, there becomes an additional possible explanation for the observations made.
Response to the reviewer
As described on page 3, lines 101-105, the UGHs had not received any training since their retirement from competition at around 15 years of age. They were housed in individual stalls measuring 4 × 4 meters and were given access to paddocks for 2 to 4 hours daily, similar to the management of UAHs. The youngest horses in the UGH groups were two individuals, both aged 17 at the onset of the study. This means they had not undergone training and lived a sedentary lifestyle for 24 months. I believe that the training effects from their earlier careers had diminished, as VO2max was reported to decrease within 12 weeks after stall rest and turnout following the cessation of 9 months of exercise training (McKeever, K. and Lehnhard, R., 2013. Age and disuse in athletes: effects of detraining, spelling, injury, and age. The Athletic Horse: Principles and Practice of Equine Sports Medicine. Elsevier Inc., pp. 243-252). Thus, I believe that the HR and HRV adaptation in the UGH group primarily resulted from the current 12-week training programme.
Notwithstanding the authors reference to a literature that supports the use of the equipment and procedures employed in this study, it should be noted that those validations are in reference to specific measures, and each study used different procedures. This validation process is thus currently incomplete and does not cover the range of heart rate variability parameters employed by the current authors. Despite this, the authors present a very broad suite of parameters and assume their validity in the horse. The authors have included primarily supportive references in their bibliography and might be somewhat selective in their choice of findings to quote. A more circumspect interpretation of findings might be advisable, while the paper could be considerably simplified and strengthened by using only those parameters that have been reliably validated.
Response to the reviewer
I’d like to thank the reviewer's comment, and many queries have been addressed in the previous reviewer’s comments. I hope that my response to the reviewer’s queries will provide clearer information and convince the reader of the outcome of this study.

Reviewer 3 Report
Comments and Suggestions for Authors
The article "Dynamic Adaptation of Heart Rate and Autonomic Regulation During Training and Recovery Periods in Response to a 12-week Structured Exercise Programme in Untrained Adult and Geriatric Horses" investigates the effects of a 12-week exercise program on heart rate and heart rate variability in untrained adult and geriatric horses. While the topic is relevant and the study design has some positive aspects, the article suffers from several significant limitations that undermine the validity and generalizability of its conclusions. Below is a more critical and detailed analysis:
Major Methodological Issues:
a) Low Sample Size and Lack of Representativeness:
The study includes only 9 horses in each group (adults and geriatrics), and two geriatric horses were withdrawn before week 8 due to health issues.
Additionally, there is no information on how the horses were selected. Were they randomly selected? Were there specific inclusion/exclusion criteria? The lack of transparency in this regard raises questions about the representativeness of the sample.
b) Lack of a Control Group:
The study does not include a control group that did not undergo the exercise program. Without a control group, it is impossible to determine whether the observed changes in HR and HRV are specifically due to the exercise program or other factors, such as natural aging, environmental changes, or dietary variations, weather conditions...
The absence of a control group is a major methodological flaw that undermines the internal validity of the study.
c) Lack of Control Over Confounding Variables:
The study does not mention whether variables such as environmental stress, social interaction among horses, or the presence of subclinical diseases were controlled. These variables could influence HR and HRV, making the results difficult to interpret.
Furthermore, it is not specified whether the horses were exposed to similar environmental conditions (temperature, humidity, etc.) throughout the study, which could affect the results.
The question is whether the changes are a real adaptation to training, or simply a non-specific adaptation to any other collective stress.
Other issues:
L95 Nine UAHs (four geldings, four mares, and one stallion, aged 20.8 ± 3.9 years, weighing 371.1 ± 51.1 kg) were selected from the teaching farms of the Laboratory Animal Centre, Kasetsart University, Nakorn Pathom, Thailand (14°00’56.0 “N 99°57’43.9” E). Nine UGHs (five geldings and four 96 mares, aged 8.8 ± 2.0 years, weighing 306.4 ± 75.8 kg
AUG and AAG are switched, at least the age does not match
Limitations in Exercise Measurement:
Lack of Quantification of Exercise Intensity:
The study did not measure speed or distance covered during the exercise sessions. This is a significant issue, as exercise intensity is a key factor influencing HR and HRV.
Without precise measurement of exercise intensity, it is impossible to determine whether the observed changes are due to duration, intensity, or a combination of both.
Although a 54-minute exercise protocol is describedin L120, there is no information on how consistency in the protocol's application was ensured. Did all horses perform the same effort? Were there variations in speed or pace among the horses?
The lack of details on the implementation of the exercise protocol raises doubts about the replicability of the study.
Issues in Data Analysis and Interpretation:
Uncontrolled Individual Variability:
The study mentions that there was considerable variability among the horses, which could affect the results. Although statistical models were used to address this variability, there is no in-depth discussion of how this variability might influence the interpretation of the results.
Specifically, which part of the records was used for analysis… how many heartbeats or how many secs?
L143 In order to ensure the accuracy of the aligned RR intervals, the automatic artefact correction feature was configured to exclude any missing, extra, or misaligned beat detections, as well as ectopic beats, including premature ventricular contractions or other arrhythmias in the RR interval time series.
Many arrhythmias are caused by autonomic tone. Removing or leaving them is controversial. If the authors remove them, they should at least discuss it.
Oversimplified Interpretation
HRV is a complex parameter influenced by multiple factors, such as stress, diet, health status, and environment. The study assumes that changes in HRV are solely due to the exercise program, which is an oversimplification.
Additionally, there is no discussion of how other factors, such as natural aging or environmental stress, might have influenced the results.
The discussion does not delve into the physiological aspects that the study demands
Similarly, in the introduction, although much better achieved, some aspects are limited to quotes without an application to the case.
For example
L49 Exercise training serves several constructive purposes for horses, enhancing overall performance and well-being [1,2]. Its primary aims include improving or maintaining optimal sports performance, delaying fatigue, minimising injury risk, and encouraging the horse’s enthusiasm for exercise [3,4]. Aerobic training plays a particularly crucial role in positively impacting biological systems by enhancing cardiovascular function [5,6], lowering resting heart rate (HR) [7-11], and increasing the aerobic capacity of skeletal muscle
The level of work a horse requires ranges from light exercise to competitive volumes at the highest level, and this volume completely alter many of these points. The training that guides this study is more similar to non-competitive human training, where it improves quality of life in virtually every aspect. Professional training has other implications with regard to stress and injury prevention. While it is true that the supporting tissues increase in strength, many athletes and horses are injured in the process. Two horses were retired for lameness. The point is that this whole superficial approach only creates more confusion.
Issues in Results Presentation:
a) Lack of Clarity in Data:
The results are presented in tables with a large amount of data. Some graphics could help to follow the data more easily.
Ethical and Animal Welfare Issues:
Withdrawal of Geriatric Horses:
Two geriatric horses were withdrawn from the study due to health issues. This raises questions about animal welfare and whether the exercise program was too demanding for geriatric horses.
There is no mention of whether measures were taken to ensure the welfare of the horses during the study, which is a significant omission in a study involving animals.
A significant percentage of geriatric horses had to be retired due to welfare issues. This should be discussed as it affects the welfare of the animals. Given the experiment, it is not clear to me that it is so beneficial for the horses, but for two of them it seems not to have been, despite having passed a thorough admission test. An exercise that produces discomfort in a significant percentage of animals is no less than questionable.
Bute use
L107 Horses were excluded if they had health issues that could affect their welfare during the experiment. Notably, two UGHs were withdrawn from this study before data collection at week 8 due to irregular gait.
I understand that horses frequently lame, this can be a problem in older horses. Have you ever given NSAIDs for possible setbacks?
Critical Conclusion:
The study has a weak methodological design due to the lack of a control group, and poor control over confounding variables. Additionally, the interpretation of the data is oversimplified, and the study's limitations are not adequately discussed. Although the topic is relevant, the study's conclusions should be taken with caution due to these limitations.
In summary, the study provides some interesting insights into the effects of exercise on adult and geriatric horses, but its methodological and interpretative limitations undermine the validity of its conclusions. Further research with more robust designs and a more critical approach to data interpretation is needed to confirm and expand on these findings.
Lack of fluency in writing:
Some sentences are too long and complex, making it difficult to understand.
Author Response
The article "Dynamic Adaptation of Heart Rate and Autonomic Regulation During Training and Recovery Periods in Response to a 12-week Structured Exercise Programme in Untrained Adult and Geriatric Horses" investigates the effects of a 12-week exercise program on heart rate and heart rate variability in untrained adult and geriatric horses. While the topic is relevant and the study design has some positive aspects, the article suffers from several significant limitations that undermine the validity and generalizability of its conclusions. Below is a more critical and detailed analysis:
Dear Reviewer 3
First of all, I’d like to thank the reviewer’s time and dedication in reviewing my work. I’ve addressed all points and the revisions are highlighted in green within the main text. The context that already mentioned was highlighted in yellow.
Major Methodological Issues:
- a) Low Sample Size and Lack of Representativeness:
The study includes only 9 horses in each group (adults and geriatrics), and two geriatric horses were withdrawn before week 8 due to health issues.
Additionally, there is no information on how the horses were selected. Were they randomly selected? Were there specific inclusion/exclusion criteria? The lack of transparency in this regard raises questions about the representativeness of the sample.
Response to the reviewer
I would like to express my gratitude to the reviewer for this comment. This study investigated the effects of a 12-week exercise program, consisting of 54 minutes of exercise per session, on heart rate (HR) and heart rate variability (HRV) in both untrained adult and untrained geriatric horses. Data were collected at two-week intervals during the exercise sessions and the subsequent recovery periods, as noted on page 4, lines 167-169. With the chosen sample size (9 horses/group), two lungers effectively organised and supervised the exercise regimen for both groups. Additionally, studies involving sample sizes of 6 to 9 horses per group have been reported in the literature regarding exercise training in horses (https://doi.org/10.2460/ajvr.71.3.314, https://doi.org/10.1152/japplphysiol.00172.2003, https://doi.org/10.1079/ECEP200426, https://doi.org/10.2527/jas.2007-0585), with some studies utilising as few as three horses in total (https://doi.org/10.5713/ajas.2002.1348). Therefore, using nine horses per group in this research was statistically reliable, even though two horses were excluded from the analysis near the end of the study.
Concerning the criteria for horse selection, it is challenging to randomly select horses for this study due to the limited availability of untrained young horses and geriatric horses in suitable conditions for long-term exercise training. The horses that met the selection criteria (as stated on page 3, lines 107-110) were included in the study. With this selection method, I believe the horses in each group were in similar conditions, aside from the age differences, and the experiment results could be reliable.
- b) Lack of a Control Group:
The study does not include a control group that did not undergo the exercise program. Without a control group, it is impossible to determine whether the observed changes in HR and HRV are specifically due to the exercise program or other factors, such as natural aging, environmental changes, or dietary variations, weather conditions...
The absence of a control group is a major methodological flaw that undermines the internal validity of the study.
Response to the reviewer
I’d like to thank the reviewer for their comment. I would typically agree that a control group containing horses that did not undergo the exercise program would be necessary if the study measured the effect of the exercise programme on heart rate variability (HRV) while “at rest”. However, this study assessed HRV adaptation “during the exercise sessions and recovery periods”. Therefore, comparing the variables between the experimental and control groups under different conditions is not appropriate. In this study, HRV modulation at the first exercise session and the recovery period at 0 weeks was used as a control for each horse group.
- c) Lack of Control Over Confounding Variables:
The study does not mention whether variables such as environmental stress, social interaction among horses, or the presence of subclinical diseases were controlled. These variables could influence HR and HRV, making the results difficult to interpret.
Furthermore, it is not specified whether the horses were exposed to similar environmental conditions (temperature, humidity, etc.) throughout the study, which could affect the results.
The question is whether the changes are a real adaptation to training, or simply a non-specific adaptation to any other collective stress.
Response to the reviewer
As stated on page 3, lines 96-110, horses in each group were selected from the same location. The UAHs and UGHs were housed in individual stalls that measured 4 × 4 meters and were given access to paddocks for 2 to 4 hours daily. Their diet consisted of 2 kg of commercial pellets and 60 to 100 grams of trace minerals, administered three times a day, along with free access to hay from hanging pergolas and tap water in their stalls. These arrangements indicate that the horses in each group lived under similar conditions and were able to engage in proper social interaction without facing environmental stress. However, these factors are expected to have a limited impact on heart rate variability (HRV) modulation during exercise sessions and recovery. Instead, the exercise intensity and duration and horse adaptability are likely to have the most significant effects on HRV modulation throughout the exercise regimen in this study.
Regarding the concern whether the adaptation accompanies the exercise program or other collective stress. With similar management and housing, these horses were deemed to face similar stress levels at rest throughout the study period. A progressive shortening of the duration of elevated HR and reduced HRV over time, as well as gradually decreased HR during exercise in both horse groups, could provide evidence of the positive effect of a 12-week exercise programme
Other issues:
L95 Nine UAHs (four geldings, four mares, and one stallion, aged 20.8 ± 3.9 years, weighing 371.1 ± 51.1 kg) were selected from the teaching farms of the Laboratory Animal Centre, Kasetsart University, Nakorn Pathom, Thailand (14°00’56.0 “N 99°57’43.9” E). Nine UGHs (five geldings and four 96 mares, aged 8.8 ± 2.0 years, weighing 306.4 ± 75.8 kg. AUG and AAG are switched, at least the age does not match
Response to the reviewer
I sincerely apologize for the serious error regarding the age range and study groups mentioned in the Materials and Methods section. Specifically, the study included nine untrained adult horses (UAHs)—four geldings, four mares, and one stallion—aged 8.8 ± 2.0 years and weighing 306.4 ± 75.8 kg. Additionally, it comprised nine untrained geriatric horses (UGHs)—five geldings and four mares—aged 20.8 ± 3.9 years and weighing 371.1 ± 51.1 kg. I have made the necessary revisions on page 3, lines 97 and 100.
Limitations in Exercise Measurement:
Lack of Quantification of Exercise Intensity:
The study did not measure speed or distance covered during the exercise sessions. This is a significant issue, as exercise intensity is a key factor influencing HR and HRV.
Without precise measurement of exercise intensity, it is impossible to determine whether the observed changes are due to duration, intensity, or a combination of both.
Although a 54-minute exercise protocol is describedin L120, there is no information on how consistency in the protocol's application was ensured. Did all horses perform the same effort? Were there variations in speed or pace among the horses?
The lack of details on the implementation of the exercise protocol raises doubts about the replicability of the study.
Response to the reviewer
Unfortunately, we could not measure the speed and distance of the exercise sessions because we were unable to install the measuring devices on the horses during round-pen lunging. These sessions did not involve any lunging aids or riders. As a result, our study does not provide information about the average speed and distance covered by the horses. This limitation is discussed in the limitations section on page 15, lines 419-423. Despite the absence of speed data, the horses exhibited consistent exercise patterns, including walking, trotting, and cantering, for durations that kept them within a limited speed range for each gait. We assumed that each horse maintained a similar speed and duration throughout the 12-week exercise period. Additionally, the exercise was supervised by two skilled lungers responsible for controlling the horses' speed during each exercise pattern. Given this experimental setup, we observed a progressive reduction in mean heart rate (HR), along with decreased stress and sympathetic nervous system (SNS) indexes. This suggests that the horses experienced HR and heart rate variability (HRV) adaptations in response to the 12-week exercise programme.
Issues in Data Analysis and Interpretation:
Uncontrolled Individual Variability:
The study mentions that there was considerable variability among the horses, which could affect the results. Although statistical models were used to address this variability, there is no in-depth discussion of how this variability might influence the interpretation of the results.
Response to the reviewer
I would like to express my gratitude to the reviewer for their insightful comments. Research indicates that horses exhibit a significant degree of individual variation (https://doi.org/10.1007/s00114-022-01811-y, https://doi.org/10.1111/evj.13017). As a result, this individual variability is inevitable, even among horses of similar age groups. In this study, the resting heart rate (HR) before each exercise session appears to have decreased in response to a 12-week exercise program, with UAH showing a resting HR of 41.2 bpm at the beginning and 39.9 bpm after 12 weeks and UGH showing 42.8 bpm at the start and 37.0 bpm at 12 weeks. However, statistical analysis did not reveal a significant difference across the different periods. This lack of significance is likely attributed to individual variation. I have already discussed this issue on page 13, lines 348-356.
Specifically, which part of the records was used for analysis… how many heartbeats or how many secs?
Response to the reviewer
In the present study, the exercise patterns—walking, trotting, and cantering—and the duration of the exercise sessions were fixed throughout the 12-week exercise period. The RR intervals were recorded in the full periods of each exercise pattern and recovery, regardless of number of heartbeats, to analyse changes in heart rate (HR) and heart rate variability (HRV) during this fixed duration, in turn, indicating the adaptation to the exercise. For instance, the mean HR during exercises 1, 2, and 3 in the UAH group decreased over the 12 weeks. Similarly, there was a decline in HR during exercises 1 and 2 in the UGH group in response to the 12-week exercise program. This reduction in HR likely indicates the body's adaptation to the exercise regimen implemented in this study.
L143 In order to ensure the accuracy of the aligned RR intervals, the automatic artefact correction feature was configured to exclude any missing, extra, or misaligned beat detections, as well as ectopic beats, including premature ventricular contractions or other arrhythmias in the RR interval time series.
Many arrhythmias are caused by autonomic tone. Removing or leaving them is controversial. If the authors remove them, they should at least discuss it.
Response to the reviewer
I would like to express my gratitude for the reviewer’s comments. It is indeed true that certain arrhythmias, such as second-degree AV blocks, are influenced by vagal tone and are generally considered to be clinically normal. However, the presence of physiological second-degree AV blocks (AVBs) can significantly impact the results of heart rate variability (HRV) analyses in horses (https://doi.org/10.1016/j.jvc.2016.10.006). AVBs can intermittently prolong RR intervals, leading to an increase in HRV (https://doi.org/10.1111/1469-8986.3510127). Moreover, the frequency domain of HRV is particularly affected by artefacts, and the type 4 error related to incorrect RR interval identification is the most common artefact observed in horses when using automatic RR detector devices (https://doi.org/10.1017/S1755254010000024). Additionally, typical artefacts—including missing, extra, or misaligned beat detections, as well as ectopic beats such as premature ventricular contractions (PVCs) or other arrhythmias—can severely distort the results of HRV analyses. Therefore, it is essential to either correct or exclude all artefacts from the analysis (https://dx.doi.org/10.1161/%E2%80%8B01.CIR.93.5.1043). In this regard, routine artefact correction is well-recognized when using automatic RR detector devices. Consequently, I believe that further discussion on artefact correction is unnecessary in this study.
Oversimplified Interpretation
HRV is a complex parameter influenced by multiple factors, such as stress, diet, health status, and environment. The study assumes that changes in HRV are solely due to the exercise program, which is an oversimplification.
Response to the reviewer
I would like to express my gratitude for the reviewer's comments. If the horses had been subjected to a structured exercise program and measurements had been taken during a "resting state," I would agree that other factors—such as stress, diet, and health status—could influence the "resting heart rate (HR) and heart rate variability (HRV)." However, all horses involved in this study received similar management, including their diet and environment. Furthermore, the present study measured HR and HRV metrics during exercise sessions and the subsequent recovery period, when sympathetic activity is more pronounced as an acute response to the exercise regimen. A decrease in HR and an improvement in HRV may result from a reduced role of sympathetic activity, leading to a shift toward increased vagal activity as the body adapts to the effects of proper exercise (https://doi.org/10.1111/j.2042-3306.2006.tb05530.x). Please correct me if I’m wrong.
Additionally, there is no discussion of how other factors, such as natural aging or environmental stress, might have influenced the results.
Response to the reviewer
In reference to the response above, if the horses had been subjected to a structured exercise program and measurements had been taken during a "resting state," I would agree that other factors—such factors could influence the "resting heart rate (HR) and heart rate variability (HRV)." However, all horses involved in this study received similar management, including their diet and environment. Furthermore, natural ageing is proved to be the factor contributing to different modulation in HR and HRV as the adaptation of these parameters is delayed when compared to the modulation in adult horses. In fact, the present study measured HR and HRV metrics during exercise sessions and the subsequent recovery period as an acute response, when sympathetic activity is more pronounced while practising exercise programmes. A decrease in HR and an improvement in HRV may result from a reduced role of sympathetic activity during exercise, leading to a shift toward increased vagal activity as the body adapts to the effects of proper exercise (https://doi.org/10.1111/j.2042-3306.2006.tb05530.x).
The discussion does not delve into the physiological aspects that the study demands
Similarly, in the introduction, although much better achieved, some aspects are limited to quotes without an application to the case.
For example
L49 Exercise training serves several constructive purposes for horses, enhancing overall performance and well-being [1,2]. Its primary aims include improving or maintaining optimal sports performance, delaying fatigue, minimising injury risk, and encouraging the horse’s enthusiasm for exercise [3,4]. Aerobic training plays a particularly crucial role in positively impacting biological systems by enhancing cardiovascular function [5,6], lowering resting heart rate (HR) [7-11], and increasing the aerobic capacity of skeletal muscle
The level of work a horse requires ranges from light exercise to competitive volumes at the highest level, and this volume completely alter many of these points. The training that guides this study is more similar to non-competitive human training, where it improves quality of life in virtually every aspect. Professional training has other implications with regard to stress and injury prevention. While it is true that the supporting tissues increase in strength, many athletes and horses are injured in the process. Two horses were retired for lameness. The point is that this whole superficial approach only creates more confusion.
Response to the reviewer
I would like to thank the reviewer for their comment. Let me clarify this issue further. The benefits of exercise training on various physiological aspects, such as improving or maintaining optimal sports performance, delaying fatigue, minimising injury risk, enhancing cardiovascular function, and increasing the aerobic capacity of skeletal muscle, are not the objectives of this study, as these have already been documented in several literature. This study aimed to estimate the modulation of heart rate (HR) and heart rate variability (HRV), measured during exercise sessions and recovery, in response to a 12-week exercise training program. The exercise patterns and duration were standardised by the two lungers across the 12-week period, ensuring that the horses ran at quite the same speed and duration during each exercise session. The progressive decrease in HR and the shortening of the duration of reduced HRV indicated the adaptation of HR and HRV in response to this exercise program. Therefore, the discussion in this study focuses on explaining the reasons behind the changes in HR and HRV variables observed.
Additionally, two geriatric horses were removed from the study due to irregular gait, exhibiting inconsistent lameness (rated 1/5) in the left front leg of one horse and the right hind limb of the other, before the week 8 measurements. This decision was made with the consideration of animal welfare. It is important to note that these two horses followed the exercise training program and displayed normal gait following the exercise. However, after spending time in a paddock with a new companion on their non-exercise day, they showed inconsistent lameness (rated 1/5) on the subsequent exercise day. I am uncertain about the events in the paddock, but I suspect these horses may have sustained minimal injuries from kicking while playing with the new companion. They required rest in the stable and applying cold gel to the affected leg. The horses were occasionally placed in small paddocks that could accommodate only one horse each. No oral or injectable pain relievers were administered. In the text (page 3, lines 111-112), I mentioned that two horses were removed due to their irregular gait to assure the reader that their removal from the study was precautionary for welfare concerns. I hope this response addresses the question adequately, and I welcome any corrections if necessary.
Issues in Results Presentation:
- a) Lack of Clarity in Data:
The results are presented in tables with a large amount of data. Some graphics could help to follow the data more easily.
Response to the reviewer
I’d like to the reviewer for this comment. In fact, I have attempted to use graphs to illustrate the study results.
However, while the graph shows trends in the variables, it is difficult to indicate statistically significant differences. Therefore, I decided to present the study results more effectively using tables.
Ethical and Animal Welfare Issues:
Withdrawal of Geriatric Horses:
Two geriatric horses were withdrawn from the study due to health issues. This raises questions about animal welfare and whether the exercise program was too demanding for geriatric horses.
Response to the reviewer
I would like to thank the reviewer for this comment. In response to the concern regarding the welfare of geriatric horses, I must respectfully disagree. Most of the geriatric horses (7 out of 9) were able to complete the 12-week program without experiencing any injuries or health issues. The two horses that were removed from the study did so due to irregular gaits, which were not attributed to the exercise program itself. As noted earlier, these two horses exhibited inconsistent lameness (rated 1/5) in one horse's left front leg and the other's right hind limb before the week 8 measurements. This decision to remove them was made with animal welfare in mind. It is important to highlight that although these horses followed the exercise training program and initially displayed normal gaits, they showed signs of inconsistent lameness after spending non-exercise days in a paddock with a new companion. I suspect that they may have sustained minor injuries from playful kicking with each other. After the incident, they required rest in the stable and the application of cold gel to the affected leg. The horses were occasionally placed in small paddocks that could accommodate only one horse each, and no oral or injectable pain relievers were administered. On page 3, lines 111-112, I mentioned that the two horses were removed due to their irregular gaits to assure the reader that this precaution was taken for their welfare.
There is no mention of whether measures were taken to ensure the welfare of the horses during the study, which is a significant omission in a study involving animals.
Response to the reviewer
I’d like to thank the reviewer for this comment. I’ve added the measure to ensure the horse's welfare during the exercise: “If any horse showed signs of exhaustion, fatigue, or lameness—indicators of compromised welfare—the exercise session would be stopped” on page 3, lines 122-124.
A significant percentage of geriatric horses had to be retired due to welfare issues. This should be discussed as it affects the welfare of the animals. Given the experiment, it is not clear to me that it is so beneficial for the horses, but for two of them it seems not to have been, despite having passed a thorough admission test. An exercise that produces discomfort in a significant percentage of animals is no less than questionable.
I would like to thank the reviewer for this comment. The reason for sample removal has been addressed in the previous reviewer’s responses. Based on the HR and HRV results in Tables 2, 6, 7 and 8, the duration of increased HR and reduced HRV was shortened in geriatric horses. For example, the duration of increased HR was observed at Ex-1 to 90 minutes of recovery period at 0 wk; however, it was shortened to Ex-1 to 30 minutes at 4-12 wk (Table 2). Moreover, the HR during Ex-2 gradually reduced from 160.6 bpm at 0 wk to 130.7 bpm at 12 wk (Table 2). In addition, the duration of decreased SD was shortened from Ex-1 to 30 minutes of recovery period at 0-6 wk to Ex-1 to Ex-3 at 8-12 wk (Table 6). Accordingly, these brief results suggest the benefit of current exercise programmes on HR and HRV adaptation while retaining the welfare of geriatric horses.
Bute use
L107 Horses were excluded if they had health issues that could affect their welfare during the experiment. Notably, two UGHs were withdrawn from this study before data collection at week 8 due to irregular gait.
I understand that horses frequently lame, this can be a problem in older horses. Have you ever given NSAIDs for possible setbacks?
Response to the reviewer
As explained on page 3, lines 108-109, the horses involved in the 12-week study had not received any medical or surgical treatments for 30 days prior or during the study period. All adult horses had never been given painkillers, while the geriatric horses had no history of receiving NSAID medications since reaching retirement age (15 years). Therefore, we can confirm that all horses were trained without the influence of NSAIDs or other medications. Additionally, it is important to note that the two geriatric horses may have suffered minor injuries due to playful kicking. They only received a cold gel application on the affected leg, and no medication was administered following this incident.
Critical Conclusion:
The study has a weak methodological design due to the lack of a control group, and poor control over confounding variables. Additionally, the interpretation of the data is oversimplified, and the study's limitations are not adequately discussed. Although the topic is relevant, the study's conclusions should be taken with caution due to these limitations.
In summary, the study provides some interesting insights into the effects of exercise on adult and geriatric horses, but its methodological and interpretative limitations undermine the validity of its conclusions. Further research with more robust designs and a more critical approach to data interpretation is needed to confirm and expand on these findings.
Response to the reviewer
As explained previously, this study assessed HR and HRV adaptation “during the exercise sessions and recovery periods”. Therefore, comparing the variables between the experimental and control groups under different conditions (measuring HRV during exercise sessions in the experiment group vs measuring HRV at rest in the control group) is not appropriate. In this study, HRV modulation at the first exercise session and the recovery period at 0 weeks was used as a control for each horse group.
I hope all responses to the reviewer’s questions provide more detail about the study and convince the reader this study provides useful information and equine field.

Reviewer 4 Report
Comments and Suggestions for Authors
This is an excellent contribution to the scientific literature. The study is well designed and the paper well written.
Author Response
This is an excellent contribution to the scientific literature. The study is well designed and the paper well written
Dear Reviewer 4
I’d like to thank the reviewer’s time and dedication in reviewing my work.
Round 2
Reviewer 2 Report
Comments and Suggestions for Authors
animals-3491324-peer-review-v2
This is a revised version of a paper previously submitted for publication. A number of issues were raised by this reviewer.
The intention in highlighting part of the text in yellow in the revised manuscript is unclear - the response states that this is context already mentioned - I hope I am correct in assuming that the authors feel the necessary response to points made was in fact already in the original paper. Accordingly, I am assessing the revision on this basis, by assuming that the insertions together with highlighted original text resolves issues identified in the original manuscript.
My review presents an overall assessment, followed by some specific issues referenced by line numbers. These specific issues are meant to provide some guidance to the authors and should not be seen as items to be addressed in a revised manuscript.
All line number references below are to the revised manuscript.
The authors make the following statement in their response to the reviewer.
"In fact, there were no genuine statistical comparisons between the two groups of horses in this study. Instead, I compared the “trend of changes” in HR and HRV metrics when two groups received the same exercise regimen and measured at a similar period during exercise sessions and recovery. In the present study, HR and HRV metrics were adapted in response to both groups' 12-week structure exercise programme. Moreover, modulating changes in HR and HRV metrics in the untrained adult horse were noticed earlier (reflecting faster adaptation) than those in untrained aged horses (2-4 weeks in untrained adult horses vs. 10-12 weeks in untrained aged horses) for example. "
It isn't entirely clear what the authors are saying here or what the word "genuine" means, but the implication is that it is appropriate to compare two groups and make comments in an otherwise ostensibly scientific investigation even though the groups are not comparable because of differences in structure or treatment, and to do so without statistical analysis. This is not in fact appropriate. Beyond the issue of whether one can comment on findings, the authors continue in the same paragraph in their response to make interpretations that could be true but are not scientifically substantiated, therefore attempting to draw credibility from a study that is not credible.
The authors do not deal adequately with the limitations associated with the techniques employed and instead offer reference to the software's manual as justification rather than deal with the science or with the mathematical basis of the routines employed. They appear to be unable to confirm that all data sets were dealt with in exactly the same way.
I see no reason to change my original recommendation on this paper and see further justification for my conclusions from the authors' response.
Line 82. The authors refer here and elsewhere to exercise "improving" HR and HRV. This raises the question of what an improvement might actually be. These indices are a reflection of the response of a body system, in this case largely the heart and circulation, to an imposed load or series of loads. Presumably the authors are referring to a change in pattern or extent of response in these indices to a standardized load that reflects some change in system work capacity and/or control mechanisms. Simple reference to an improvement is not meaningful unless the context is fully defined. Additional references are needed since two of the three citations reference the author's own work.
Line 91. The authors claim in their response that they have not compared to the two groups in their analysis yet here they claim to be assessing recovery in groups performing in the same structured program. It is acknowledged in their response to the reviewer that the two groups were exercised under different conditions, separately. Comparison is in fact explicit in the very experimental design they describe, yet here they describe the structured programs as being the same.
Line 97. "… Were selected from…". How? These details are supposedly in the highlighted section starting on line 101 - was the selection of equal group sizes part of the process, and if so, were the selected horses selected at random from the group that met the inclusion criteria are were horses meeting the inclusion criteria selected until the required group of nine horses had been filled?
Line 116. "… Horses were stabled." Does this mean they did not receive daily turnout during the study on exercise days?
Line 115. The authors' response indicates that conditions for the two different groups were not identical - both facilities and staff were different. The authors cannot separate influence of location from that of group (age and exercise history).
Line 138. The authors' reference to diurnal variation does not justify the failure to randomize the order of treatment of horses but rather it reinforces the need for the order to be randomized and for this random variable to be included in the statistical analysis.
Line 171. Changes here are meant to clarify the issue of group versus time interaction, but they don't. The word "group" here appears to mean the group of horses at each time point within a training session, and these are described as independent, but they clearly are not. Time is described as referring to the 12 week periods but here N = 1 unless the authors are referring to the two different overall study groups of UAH and UGH between which there is supposed to be no formal statistical comparison. Elsewhere (Table 2), "group" does refer to these two groups.
An issue not raised in the previous review but raised here for the sake of completeness is that of the authors use of the word "adaptation". What the author is describing is a training response. The word adaptation strictly refers to the evolutionary process by which natural selection favours traits most suited to the species' environment and challenges, and is a process that takes place over a much longer timescale than is relevant to the training of an individual. Individual animals respond to environmental pressures within the limits of their ability to do so, those limits being imposed by their phenotype.
Comments on the Quality of English Language
The authors' technical use of English is very good, as is their grammar, but there is an odd issue with terminology, its use and consistency. This may not be an issue with use of language, but I do not have the technical expertise to analyse the issue further.